# Synaptic organization of the *Drosophila* antennal lobe and its regulation by the Teneurins

**Timothy J Mosca\*, Liqun Luo\***

Department of Biology, Howard Hughes Medical Institute, Stanford University, Stanford, United States

**Abstract** Understanding information flow through neuronal circuits requires knowledge of their synaptic organization. In this study, we utilized fluorescent pre- and postsynaptic markers to map synaptic organization in the *Drosophila* antennal lobe, the first olfactory processing center. Olfactory receptor neurons (ORNs) produce a constant synaptic density across different glomeruli. Each ORN within a class contributes nearly identical active zone number. Active zones from ORNs, projection neurons (PNs), and local interneurons have distinct subglomerular and subcellular distributions. The correct number of ORN active zones and PN acetylcholine receptor clusters requires the Teneurins, conserved transmembrane proteins involved in neuromuscular synapse organization and synaptic partner matching. Ten-a acts in ORNs to organize presynaptic active zones via the spectrin cytoskeleton. Ten-m acts in PNs autonomously to regulate acetylcholine receptor cluster number and transsynaptically to regulate ORN active zone number. These studies advanced our ability to assess synaptic architecture in complex CNS circuits and their underlying molecular mechanisms.

## Introduction

Understanding how neural circuits process information requires knowing not only the neuroanatomical connectivity and electrophysiological properties of individual neurons, but also the organization of synapses between specific neuronal types. Knowledge of these basic organizational principles, as well as the mechanisms for how they are achieved in a complex circuit, represents a critical goal for neuroscience. Historically, neuromuscular junctions have been model synapses for mechanistic studies because of their ease of access (*Keshishian et al., 1996*; *Sanes and Lichtman, 1999*). However, neuromuscular circuits display less complexity compared to neural circuits in the central nervous system (CNS). Serial-section electron microscopy (EM) has offered remarkable resolution in studying CNS synapses in the context of circuits, including the entire Caenorhabditis *elegans* nervous system (*White et al., 1986*), the mouse olfactory bulb (*Hinds and Hinds, 1976a*, *1976b*), retina (*Briggman et al., 2011*; *Helmstaedter et al., 2013*), and visual cortex (*Bock et al., 2011*), and the *Drosophila* visual system (*Meinertzhagen and O'Neil, 1991*; *Takemura et al., 2013*). However, these approaches are labor intensive and not easily amenable to analysis following genetic perturbation, at varying developmental stages, or with intact samples. Indeed, even for well-known synaptic organizers like Neurexin and Neuroligin (*Chia et al., 2013*), studies investigating their function were rarely carried out in an intact system with a resolution sufficient to reveal their detailed functions in synaptogenesis between defined neuronal types. As such, there is a need to examine CNS synapses in the intact brain, at the level of light microscopy in a system with sufficient complexity to reflect features of the CNS, but tractable enough to dissect connectivity and function using genetic tools. Current work in *Drosophila* has sought to fulfill this need (*Kremer et al., 2010*; *Christiansen et al., 2011*; *Berger-Muller et al., 2013*; *Chen et al., 2014*).

**\*For correspondence:** tmosca@stanford.edu (TJM); lluo@stanford.edu (LL)

**eLife digest** Just as progress in science relies on researchers communicating their findings to other people working in their field, our bodies rely on neurons being able to communicate with other neurons. This is where structures called synapses come in: synapses allow signals to be passed from one neuron to another. Neurons and synapses process information by forming circuits in the brain, but relatively little is known about how synapses develop or how they are organized within circuits.

Mosca and Luo have now examined a neural circuit in the fruit fly (*Drosophila*) that receives sensory information about smells in the environment, and then converts this information to signals which can be understood by other parts of the brain. This particular circuit has previously been identified as a good model of how the brain processes information.

Mosca and Luo found that the synapses in this circuit were organized according to specific 'rules' that determined factors such as the quantity and location of synapses at different points in the circuit. Additionally, it was found that the successful development of synapses required the involvement of two members of a family of proteins called the Teneurins: this family of proteins is involved in a variety of neurodevelopmental processes.

Teneurins have been implicated in bipolar disorder, and malfunctioning synapses are thought to be associated with a number of other mental health conditions, so the results of Mosca and Luo could lead to a better understanding of these conditions.

The *Drosophila* olfactory system contains complex circuits that transform chemosensory input to the regulation of behavioral output. Olfactory information is received by first-order olfactory receptor neurons (ORNs) and is transmitted by ORN axons to the antennal lobe, the first olfactory processing center in the brain. There, the information is conveyed to the dendrites of second-order projection neurons (PNs) through class-specific ORN–PN synapses. PN axons carry signals to higher-order brain centers (*Vosshall and Stocker, 2007*). The local interneurons (LNs) innervate the antennal lobe extensively with their dendrites (*Chou et al., 2010*) and are involved in transforming ORN input to PN output (*Wilson, 2013*). These three groups of neurons have extensive interconnectivity and indeed, the antennal lobe has emerged as a model neural circuit for investigating the general principles of information processing (*Wang, 2012*; *Wilson, 2013*; *Twick et al., 2014*). While many antennal lobe neurons have been examined electrophysiologically (*Wilson, 2013*), there has been little analysis of their synaptic architecture: how many connections are made, where they are made in relation to other neurons in the circuit, and what molecules are required for their organization. Electron microscopy has been used to probe different genetic conditions (*Acebes and Ferrus, 2001*; *Acebes et al., 2011*) but could not definitively assign glomerular identity or cell type for individual synapses, highlighting the need for structural analysis at the level of light microscopy in three dimensions. Knowledge about synaptic organization is critical to understand how sensory information is received, processed, and relayed by this circuit. Thus, the fly olfactory circuit represents an excellent opportunity to assess synaptic organization in an intact system at the resolution of individual, defined neuronal classes, linking structure to physiology and function.

Here, we utilized an approach where presynaptic active zone and postsynaptic neurotransmitter receptor proteins were tagged with fluorescent molecules to highlight and quantitatively describe the synaptic organization of the *Drosophila* antennal lobe. We identified distinct rules that govern the number, density, and spatial organization of olfactory synapses for multiple classes of neurons. To understand how these rules are implemented at the molecular level, we investigated the function of the Teneurins, evolutionarily conserved type II transmembrane proteins recently shown to regulate synaptic partner matching and neuromuscular synaptic organization (*Hong et al., 2012*; *Mosca et al., 2012*). We show that presynaptic Ten-a and postsynaptic Ten-m regulate central synapse number and active zone structure. These represent the first in vivo functions of Teneurins in organizing CNS synapses, which could contribute to their association with brain disorders such as bipolar disorder.

## Results

### Labeling presynaptic active zones using Bruchpilot-Short

To visualize synapses in vivo in genetically identifiable cells, we explored strategies to label active zones, sites of presynaptic neurotransmitter release (*Zhai and Bellen, 2004*). In *Drosophila*, the primary unit of active zone architecture is an electron dense structure called the T-bar, a specialized formation that promotes vesicle docking and release (*Wichmann and Sigrist, 2010*) present at peripheral (*Jia et al., 1993*) and central synapses (*Meinertzhagen and O'Neil, 1991*; *Prokop and Meinertzhagen, 2006*). The major structural components of the T-bar include Bruchpilot (Brp) and Rim-binding protein (*Wagh et al., 2006*; *Liu et al., 2011*), with Brp encoding the 'table-top' of the T-bar and Rim-binding protein comprising the 'pedestal' (*Figure 1A*). Widely used monoclonal antibodies (nc82) against Brp (*Blanco Redondo et al., 2013*) label synaptic neuropil and discrete puncta corresponding to active zones (*Wagh et al., 2006*). However, these antibodies cannot distinguish synapses formed by a given class of neurons in complex CNS circuits. Therefore, to visualize active zones in genetically identifiable cells, we sought to label Brp with the GAL4/UAS system (*Brand and Perrimon, 1993*).

Full-length Brp is a 1740 amino acid protein that can aggregate upon ectopic overexpression (*Wagh et al., 2006*), limiting its use as a labeling strategy. However, Bruchpilot-Short (Brp-Short), a nonfunctional 754-residue portion of Brp (*Figure 1A*), localizes to endogenous sites of full-length Brp (*Schmid et al., 2008*; *Fouquet et al., 2009*) without deleterious effects. Thus, under the control of a binary expression system, Brp-Short can act as a reporter of endogenous active zones without disrupting morphology or function (*Schmid et al., 2008*; *Fouquet et al., 2009*; *Kremer et al., 2010*). This strategy has been employed at neuromuscular synapses (*Schmid et al., 2008*; *Fouquet et al., 2009*), in the mushroom body (*Kremer et al., 2010*; *Christiansen et al., 2011*), and in the visual system (*Berger-Muller et al., 2013*).

We utilized Brp-Short to quantitatively measure the number of active zones, a key element of understanding synaptic organization. We expressed an mStrawberry-tagged version (Brp-Short-mStraw: *Owald et al., 2010*) in Or67d ORNs that innervate the DA1 glomerulus. Coexpression with a known presynaptic protein, DSyd-1, revealed extensive colocalization (*Figure 1—figure supplement 1*, panel A), suggesting Brp-Short-mStraw correctly localizes to synapses. Further, RNAi against a portion of endogenous Brp not encoded by the Brp-Short transgene in those ORNs resulted in a loss of Brp-Short signal at ORN presynapses (*Figure 1—figure supplement 1*, panels B–C), though Brp-Short was still detectable in the cell body. This supports the hypothesis that Brp-Short requires endogenous Brp for proper localization to synapses; alternatively, Brp-Short could require endogenous Brp for proper trafficking to axon terminals. Finally, to confirm the veracity of the synaptic signal, we performed immuno-electron microscopy (immuno-EM) using antibodies to the GFP-tag (the mStraw tag is not amenable to immuno-EM) of a Brp-Short-GFP transgene (*Schmid et al., 2008*) expressed in all ORNs via *pebbled-GAL4* (*Sweeney et al., 2007*). With *pebbled-GAL4*, Brp-Short expression and localization was similar to that observed for ORN-class-specific GAL4 drivers (data not shown). While immuno-EM cannot be utilized as a quantitative strategy for determining the number of active zones (as the conditions needed to preserve the GFP epitope necessarily reduce the resolution of the technique to clearly discern all synaptic densities), it can be used to verify that Brp-Short localizes to electron-dense regions that are consistent with synaptic active zones. Indeed, 5-nm gold particles were observed to decorate putative active zone profiles (*Figure 1—figure supplement 1*, panel D) in ultrathin sections of fly antennal lobes. 90.1% of Nano-Gold dots colocalized with synaptic densities (675 5-nm Nano-Gold particles out of 749 particles observed). Combined, these experiments demonstrate that Brp-Short localization is consistent with active zones in the fly CNS.

To determine the number of active zones produced by ORN axons that project to a given glomerulus, we expressed Brp-Short-mStraw in various ORN classes (*Figure 1B–D*, *Figure 1—figure supplement 2*) together with *UAS-mCD8-GFP* (*Lee and Luo, 1999*) as a general neurite label. Single optical sections (*Figure 1B–D*, insets) revealed the punctate nature of the Brp-Short-mStraw signal. To quantify the number of Brp-Short puncta in these ORNs, we imaged individual glomeruli using high magnification confocal microscopy followed by image deconvolution and 3-D rendering (*Video 1*; see 'Materials and methods'). Image processing allowed us to calculate the number of Brp-Short puncta (*Figure 1B,E*) as 'Spots' and the neurite volume by surface rendering of the membrane marker mCD8-GFP (*Figure 1C,F*). The numbers of Brp-Short puncta exhibited low variability for multiple classes of ORNs (*Figure 1H*, *Figure 1—figure supplement 2*), suggesting that neuronal classes form stereotyped

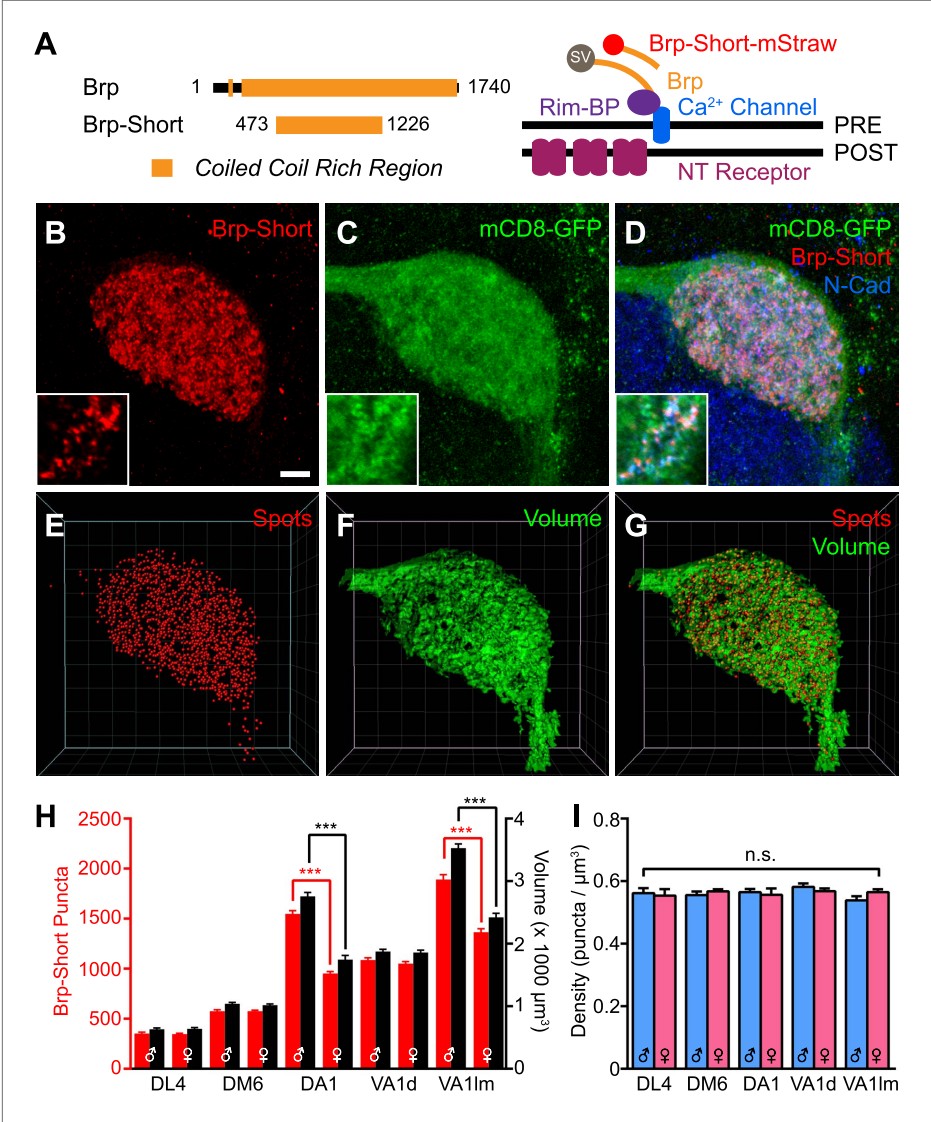

**Figure 1**. Measuring CNS synapses using Bruchpilot-Short. (**A**) Diagram comparing full-length Bruchpilot and Bruchpilot-Short. Numbers denote amino acids and orange represents coiled coil regions (after *Wagh et al., 2006*). The positions of Brp and Brp-Short in relation to known active zone proteins are also shown in a simplified format (after *Liu et al., 2011*). (**B**–**D**) High magnification confocal stacks of a single DA1 glomerulus with putative synapses labeled by Brp-Short (**B**), neurites labeled by mCD8-GFP (**C**), and the merge (**D**). Insets show higher magnification of a single optical section to demonstrate the punctate nature of Brp-Short-mStraw. (**E**–**G**) Screenshots of three-dimensional renderings show conversion of Brp-Short into 'Spots' (**E**) and mCD8-GFP into a volumetric surface rendering (**F**). For (**F**) and (**G**), the transparency of the surface rendering is at 60% to highlight internal structure and make Brp-Short puncta visible. (**H**) Quantification of Brp-Short puncta (red, left axis) and neurite volume (black, right axis) in males and females for five ORN classes. Statistical comparisons between males and females of a single genotype were done by student's *t* test. Significance across all genotypes for density was assessed with a one-way ANOVA and corrected for multiple comparisons by a posthoc Tukey's multiple comparisons test. ***$p < 0.001$. In all cases, $n \geq 9$ brains, 18 antennal lobes for each genotype. (**I**) Quantification of synaptic density (Brp-Short puncta/volume of neurites) in males (blue) and females (pink) for five ORN classes. Despite considerable differences in Brp-Short puncta number, all classes of ORNs display nearly identical synaptic densities. Scale bar = 5 µm for all images.

The following figure supplements are available for figure 1:

**Figure supplement 1**. Validation of Brp-Short.

**Figure supplement 2**. Additional examples of the Brp-Short assay.

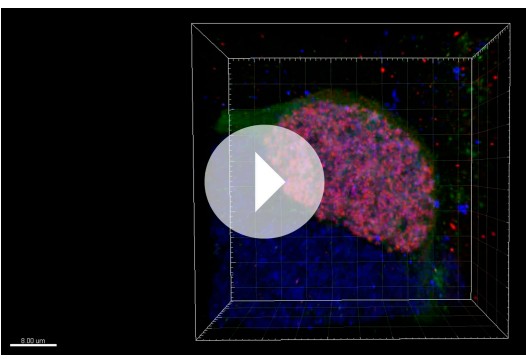

**Video 1**. Three-dimensional rendering of a Brp-Short and mCD8-GFP in a DA1 glomerulus. (related to **Figure 1**) Animation of a three-dimensional rendering of the glomerulus shown in **Figure 1B–D**. Each channel is highlighted and the transformation to either Spots (for Brp-Short puncta) or a surface rendering to measure neurite volume (for mCD8-GFP staining) is shown. High magnification of the rendered glomerulus reveals the distribution of Brp-Short puncta within the glomerulus.

numbers of active zones across animals. While extensive EM reconstruction is not available for the antennal lobe (and thus, absolute counts of olfactory synapses in each glomerulus for comparison), the recent use of labeled Brp in the *Drosophila* visual system indicated that one Brp punctum is equivalent to one active zone (**Chen et al., 2014**). Therefore, it is likely that the Brp-Short label largely reflects actual active zone number.

In examining five classes of ORNs, we found that ORN synapse number, as indicated by the number of Brp-Short puncta, scaled with ORN axon volume (**Figure 1H**), which itself scaled with glomerular size. Moreover, the assay revealed sex-specific differences in synapse number for two glomeruli, DA1 and VA1lm, that coincide with known sex-specific differences in glomerular volume (**Stockinger et al., 2005**). Remarkably, despite differences in synapse number and ORN neurite volume, all observed ORN classes had a constant synaptic density of 0.56 Brp-Short puncta per $\mu m^3$ (**Figure 1I**). These data suggested the existence of an organizational mechanism that controls synaptic density. In all, our results validated the use of Brp-Short as a quantifiable presynaptic active zone label in central neurons and revealed quantitative parameters that govern the synaptic contacts of ORNs: a stereotyped number of connections for each neuronal class from animal to animal and a constant synaptic density regardless of ORN class.

## Each ORN contributes a discrete, nearly identical number of synapses to a glomerulus

In the olfactory system, individual ORNs of a particular class (as denoted by their odorant receptor expression) project to an identifiable glomerulus (**Vosshall and Stocker, 2007**). In terms of their projection pattern (**Jefferis and Hummel, 2006**) and physiological function (**Wilson, 2013**), individual ORNs can largely be treated equivalently. However, it is unknown whether each neuron contributes an equal number of synapses to the aggregate, stereotyped average or whether there is heterogeneity in synapse number at the level of individual neurons. Different results would suggest distinct rules for how individual neurons determine how many connections to form. To address this, we utilized MARCM (**Lee and Luo, 1999**) to label the synapses and membranes of individual (or small subsets of) ORNs that innervate two glomeruli, DL4 and DM6.

Most frequently, we obtained 2–3 cell clones (**Figure 2A–B**), but we observed occurrences of single-cell clones (**Figure 2—figure supplement 1**, panels A–B) and large clones (**Figure 2—figure supplement 1**, panels C–D) that encompassed the entire glomerulus. Single neuron clones were readily discernible in confocal stacks; cell numbers could not be precisely assigned to larger than 2–3 cell clones, however, due to axon fasciculation. For all clones (*n* = 100 for DL4, 99 for DM6) except the largest 7 for DL4 and 9 for DM6, we quantified the number of Brp-Short puncta and the volume of the mCD8-GFP-labeled neurites as above (**Figure 1**), and plotted the distribution of puncta number across all clones in histograms. For both DL4 (**Figure 2C**) and DM6 (**Figure 2D**), we observed that as clone size increased, the number of Brp-Short puncta appeared to increase in discrete steps of a similar size, suggesting that each individual neuron contributes an equal number of synapses to the glomerular average. If each neuron contributed a widely different number of synapses to the average, we would have observed a more continuous distribution.

To statistically analyze the distributions for DL4 and DM6 ORNs, we determined their best-fit relationships using the first four peaks for both data sets, which had the most observations. Each individual peak could be reliably fit using a single Gaussian relationship ($r^2$ > 0.98, **Figure 2C–D**, colored dashed lines), and the entire set of peaks could be fit by a sum of Gaussians (see **Figure 2—source data 1**;

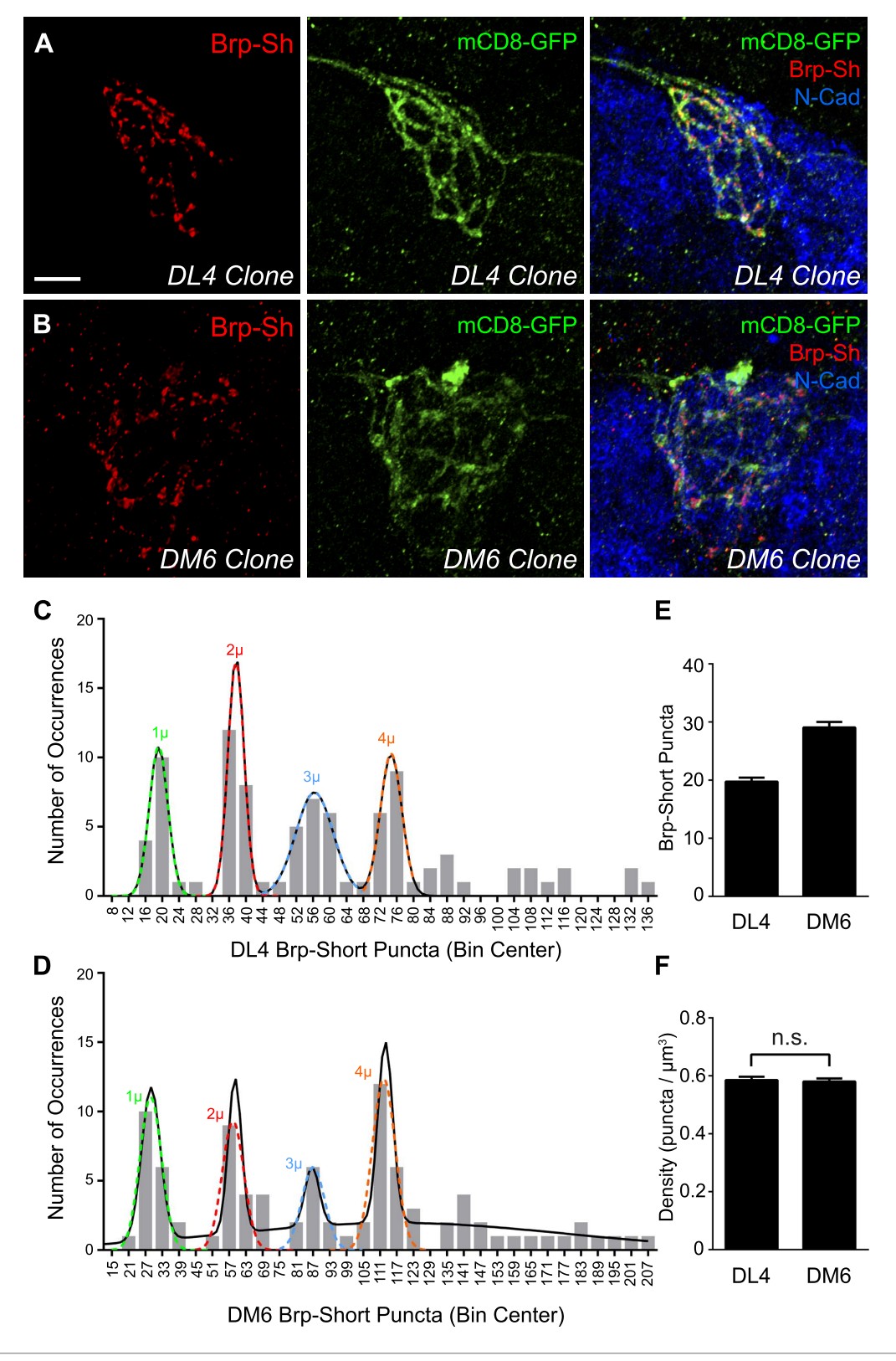

**Figure 2**. Clonal analysis of ORN synapse number. Representative high magnification confocal stacks of small clones of DL4 ORNs (**A**) and DM6 ORNs (**B**) expressing Brp-Short-mStraw (red) and mCD8-GFP (green). (**C**) Frequency histogram of Brp-Short puncta in DL4 clones (*n* = 93 clones from 64 animals). (**D**) Frequency histogram of Brp-Short

*Figure 2. Continued on next page*

*Figure 2. Continued*

puncta in DM6 clones (*n* = 90 clones from 64 animals). Both graphs exhibit notable periodicity, suggesting that each neuron contributes approximately the same number of synapses. Colored dashed lines indicate Gaussian fits for individual peaks. Solid black lines represent the best-fit sum of Gaussian relationship for each dataset. (**E**) Quantification of Brp-Short puncta from identified single neuron clones to DL4 (*n* = 16 clones) or DM6 (*n* = 17 clones). (**F**) Quantification of synapse density from all clones. In both DL4 and DM6, the clonal average density is identical to the class average density. In (**E**) and (**F**), data represent mean ± SEM. Scale bar = 5 μm.

The following source data and figure supplements are available for figure 2:

**Source data 1**. Table of Gaussian curve-fitting data.
**Source data 2**. Table of resampling approach.
**Figure supplement 1**. Additional DL4 and DM6 ORN MARCM clones.

---

'Materials and methods'). The DL4 relationship was best fit by a 4-term sum of Gaussians (*Figure 2C*, solid black line) with a value of 19.13 for the apex of the first peak, corresponding to the 'single cell' contribution of active zone puncta. This is in close agreement with our original data of 19.75 ± 0.68 puncta for single cell clones (*Figure 2E*). The DL6 relationship was best fit by a 5-term sum of Gaussians (*Figure 2D*, solid black line), with a value of 28.62 active zone puncta as the 'single cell' contribution, which is again in strong agreement with our original data of 29.06 ± 0.95 puncta for single cell clones (*Figure 2E*). Thus, in both cases, a sum of Gaussians can accurately represent the observed peaks and predict single 'quantum' contributions similar to those observed for single cell clones. These 'quantum' values are consistent with the 'step size' of Brp-Short puncta measured in clones with increasing size (*Figure 2C–E*). Further, in comparing DL4 and DM6, the difference in the number of puncta per single cell clone (*Figure 2E*) is consistent with the numerical relationship between the aggregate glomerular averages (*Figure 1H*). All observed clones also displayed a nearly identical synaptic density (*Figure 2F*) regardless of their size, consistent with measurements in whole glomeruli.

To further test whether each ORN contributes the same number of synapses to the aggregate, we examined the spacing of the peaks. If the peaks were perfectly evenly spaced, the standard deviation in the distances (S) between each peak would be 0. Using the individual Gaussian relationships (colored dashed lines), we calculated an S value of 0.235 for DL4 and 2.26 for DM6. While visual assessment suggested uniformity, we used these values to more rigorously assess this. We used a resampling approach and calculated an S value for each simulation to test the null hypothesis that there is no periodic relationship between the peaks. A lower S value than that which was observed means that the simulation resulted in peaks that were more evenly spaced. *Figure 2—source data 2* details these results. For distributions of DL4 or DM6, there was a low probability (<0.00048 or <0.05, respectively) of seeing peaks as evenly distributed as that observed. Thus, we reject the null hypothesis of no quantal relationship for both DL4 and DM6. In all, this approach demonstrates that the peaks have equal spacing in between, further supporting our assertion that each ORN contributes an equivalent number of active zones to the aggregate and that each class of ORNs has a characteristic 'step size' consistent with its aggregate average.

## Brp-Short reveals subglomerular synaptic architecture in olfactory neurons

To better understand CNS synaptic organization, it is important to know not only how many connections are formed, but also where they form. Brp-Short allowed us to examine the spatial organization of active zones of any specific neuronal class within individual glomeruli. As an approximation of the complement of active zones for the DA1 glomerulus, we examined its component ORNs, PNs, and LNs. Using GAL4 drivers for the ORNs (*Or67d-GAL4*), PNs (*Mz19-GAL4*), and an LN line that includes most ipsilateral LNs (*NP3056-GAL4*), all of which innervate the DA1 glomerulus, we examined the localization of their synapses. For all classes, Brp-Short puncta colocalized with endogenous Brp staining (*Figure 3—figure supplement 1*, panels A–C) recognized by an antibody whose epitope is not contained within Brp-Short (*Wagh et al., 2006*), demonstrating that Brp-Short recognizes endogenous active zones. These three classes of neurons, when combined, provide 3342 ± 99 Brp-Short puncta per

glomerulus. When the endogenous number of active zones is quantified using an antibody to endogenous Brp using the same image-processing strategy, DA1 contains 3849 ± 80 active zones (*Figure 3—figure supplement 1D*). This suggests that the majority of synapses in the DA1 glomerulus are indeed made by *Or67d-GAL4*+ ORNs, *Mz19-GAL4*+ PNs, and *NP3056-GAL4*+ LNs; the discrepancy may be contributed by synapses made by additional neurons, such as other classes of local interneurons (*Chou et al., 2010*), atypical projection neurons (*Lai et al., 2008*), or potentially peptidergic neurons (*Ignell et al., 2009*; *Carlsson et al., 2010*), or may represent a slight underestimation by the Brp-Short assay.

Using single optical sections within the DA1 glomerulus, we observed two levels of organization: (1) the location of neurites and Brp-Short puncta with respect to the glomerulus (the subglomerular distribution), and (2) Brp-Short puncta with respect to their own neurites for each class of neurons (the subcellular distribution). Of the three classes of neurons observed, ORNs contribute the majority of the Brp-Short puncta (1632 ± 22) and were widely distributed within the glomerulus (*Figure 3A,D*), though with distinct voids (*Figure 3D*, asterisk) where the glomerulus lacked ORN neurites and Brp-Short puncta. PNs were the second major contributors of Brp-Short puncta within the glomerulus (1182 ± 24). These connections likely synapse onto PN and LN dendrites, as shown by electrophysiology (*Kazama and Wilson, 2009*), and are analogous to the dendrodendritic synapses between secondary dendrites of mitral cells and olfactory bulb interneurons in the vertebrate olfactory bulb (*Urban and Arevian, 2009*). These Brp-Short puncta showed a more even distribution throughout the glomerulus (*Figure 3B,E*), with smaller voids than those observed for ORNs despite reduced overall density. Finally, local interneurons (LNs) contribute the fewest Brp-Short puncta (528 ± 54) of the three classes. The LNs accessed by the *NP3056-GAL4* line make up the majority of ipsilateral LNs including most or all pan-glomerular projections (*Chou et al., 2010*), but may not be representative of all classes of LNs. Within the glomerulus, LN distribution was the most restricted (*Figure 3C,F*). When examined at equivalent optical sections, these projections mostly filled the gaps in the ORN projection pattern, consistent with previously observed results for neurites (*Hummel and Zipursky, 2004*).

With respect to their own neurites, Brp-Short puncta in ORNs were not evenly distributed: distinct sections of neurite lack Brp-Short puncta (*Figure 3D*, arrow) while other sections had large clusters (*Figure 3D*, arrowheads) of Brp-Short puncta (*Figure 3D*). To quantify this distribution, we conducted a nearest neighbor distance (NND) analysis for Brp-Short puncta in each class of neurons, which has been informative in understanding spatial relationships between synapses (*Bleckert et al., 2013*). We found that all three types of neurons had distinct mean NND values (*Figure 3G–I*): ORNs had the closest puncta and LNs the furthest with PNs falling in between the two. To quantify the clusters observed, we defined clustered puncta as having an NND of 0.75 µm or less (an NND of 0.6 µm corresponds to directly touching). For ORNs, these large clusters amounted to 17% of puncta that fell within this range (*Figure 3G*). PNs lacked these large clusters, but had some smaller clusters, as well as extended regions of neuronal membrane devoid of Brp-Short puncta (*Figure 3E*). Indeed, only 7.9% of PN puncta were clustered by this metric (*Figure 3H*). Within LN neurites, Brp-Short puncta also displayed significant clustering (15%; *Figure 3I*). However, unlike ORNs and PNs, no extended regions of membrane were observed without synaptic puncta. This suggests a more even distribution for the remaining puncta, which is supported by the higher value for NND (*Figure 3I*) and consistent with previous imaging using a synaptic vesicle marker (*Chou et al., 2010*). Overall, these results show distinct synaptic architecture, with respect to the glomerulus and the neurites themselves, made by three different types of olfactory neurons. Moreover, distinct parameters can be defined for each class in terms of how puncta are clustered and the minimum distance between puncta. Though some distances may be underestimated, these facets may represent additional rules that govern synaptic organization in the antennal lobe.

## Teneurin-a is required presynaptically for normal ORN synapse number

Having utilized Brp-Short to determine aspects of presynaptic organization in olfactory neurons (*Figures 1–3*), we next sought to investigate the molecular mechanisms that may underlie these rules. We began with genetic perturbations of candidate molecules, the Teneurins. Teneurins are type II transmembrane proteins involved in neuronal development (*Young and Leamey, 2009*) and are disrupted in human neurological disorders (*Aldahmesh et al., 2012*; *Psychiatric GWAS Consortium Bipolar Disorder Working Group, 2011*). Recently, we demonstrated that the *Drosophila* Teneurins, Ten-m and Ten-a, regulate synaptic partner matching in the olfactory and neuromuscular systems via homophilic interaction between elevated Teneurin levels (*Hong et al., 2012*; *Mosca et al., 2012*).

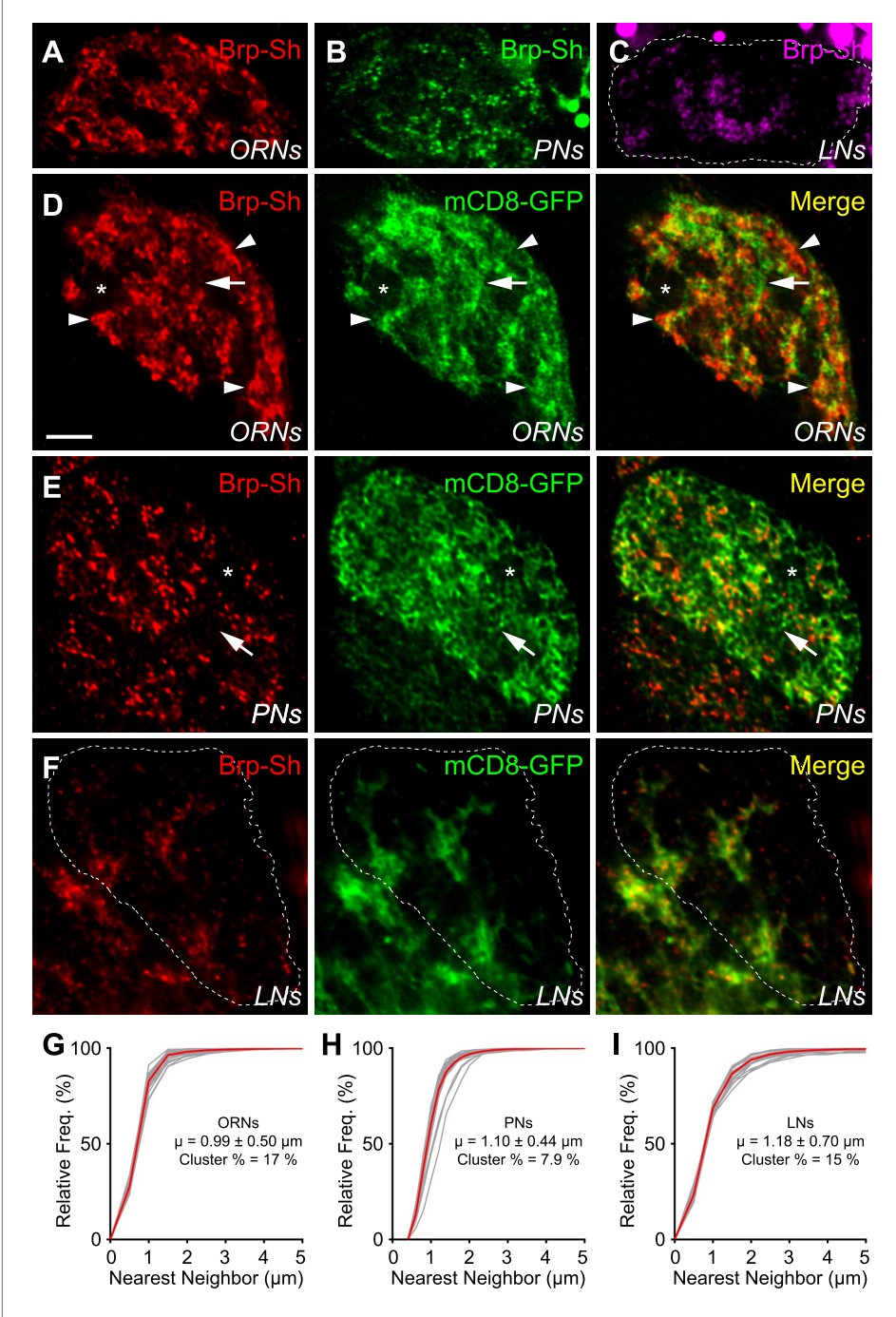

**Figure 3**. Antennal lobe neurons display distinct subglomerular architecture. (**A**–**C**) Single, high-magnification confocal optical sections through the center of the DA1 glomerulus in three animals where ORNs (**A**), PNs (**B**), or LNs (**C**) are expressing Brp-Short. A different color denotes the synapses of each class. (**D**) Single, high magnification confocal optical sections through the center of the DA1 glomerulus where ORNs express Brp-Short-mStraw (red) and mCD8-GFP (green). Brp-Short clusters (arrowheads) are visible, along with regions of neurite devoid of Brp-Short puncta (arrow) and voids where ORNs do not project (asterisk). (**E**) Single, high magnification confocal optical sections through the center of the DA1 glomerulus where PNs are expressing Brp-Short-mStraw (red) and mCD8-GFP (green). Regions of PN neurite without Brp-Short puncta are visible (arrow) as well as small voids lacking Brp-Short puncta and neurite projections (asterisk). (**F**) Single, high magnification confocal optical sections through the center of the DA1 glomerulus where LNs are expressing Brp-Short-mStraw (red) and mCD8-GFP (green). Brp-Short puncta are
*Figure 3. Continued on next page*

*Figure 3. Continued*

largely distributed evenly throughout neurites. Images in panels **D**–**F** (from three animals) demonstrate distinct patterns of synapse localization and distribution within the glomerulus. Dashed lines denote the glomerulus in (**C**) and (**F**) as defined by N-Cad staining (not shown). Scale bar = 5 μm. (**G**–**I**) Cumulative frequency histograms of the nearest neighbor distance between Brp-Short puncta in ORNs (**G**), PNs (**H**), and LNs (**I**). The average is indicated (μ) and the Cluster % of puncta with an NND ≤ 0.75 μm. Gray traces represent individual glomeruli. Red traces represent the aggregate average. In all cases, $n \geq 10$ animals, 19 antennal lobes, and 400 (**I**), 800 (**H**), or 1400 (**G**) individual puncta.

The following figure supplement is available for figure 3:

**Figure supplement 1**. Comparison of Brp-Short to endogenous Brp.

Basal levels of the Teneurins interact heterophilically at NMJ synapses to organize connections (***Mosca et al., 2012***). However, due to the absence of techniques for quantitatively examining central synapses in vivo, a role for the Teneurins in organizing central synapses has not been established. Due to the similarity in partner matching mechanisms between central and peripheral nervous systems, and the fact that both Ten-a and Ten-m are expressed at basal levels in all glomeruli (***Hong et al., 2012***), we hypothesized that the CNS may use Teneurins similarly to the PNS for synaptic organization. Specifically, we hypothesized that olfactory neurons would utilize basal levels of Ten-a and Ten-m to control synaptic number.

To test this hypothesis, we used Brp-Short to quantify active zone number in Or47b-positive ORNs innervating the VA1lm glomerulus. Ten-a is expressed at a basal level in this glomerulus, and therefore does not directly affect synaptic partner matching (***Hong et al., 2012***). Comparing wild-type (***Figure 4A***) and *ten-a* null mutant flies (***Figure 4B***), we found that Or47b ORNs in the *ten-a* mutant (***Figure 4B***) had 23% fewer Brp-Short puncta (***Figure 4D***). Notably, loss of *ten-a* did not alter ORN axon volume (***Figure 4E***), despite affecting glomerular morphology (***Figure 4B***), which is due to the loss of *ten-a* in neurons of the nearby DA1 glomerulus (***Hong et al., 2012***). As such, synaptic density was similarly reduced (***Figure 4F***). To ensure that changes in glomerular morphology were not related to changes in synapse number, we also examined ORNs that innervate the DL4 glomerulus (***Figure 4—figure supplement 1***). In *ten-a* mutants, glomerular morphology was unchanged in 87% of animals examined, but all animals displayed a similar reduction in synapse number and density without affecting the axon volume (***Figure 4—figure supplement 1***, panels C–E). Interestingly, at the NMJ, Teneurin perturbations reduced active zone and synaptic bouton numbers, but did not alter active zone density (TM, unpublished observations). However, in ORNs, neurite volume was unchanged, so active zone density was reduced, suggesting a difference in the specific consequence of Teneurin perturbation in the CNS compared with the NMJ.

As *ten-a* mutants lack Ten-a in ORNs and PNs, we next sought to determine where Ten-a was required to regulate normal synapse number using tissue-specific rescue and RNAi experiments. Restoring Ten-a expression to the Or47b ORNs in the *ten-a* mutant (***Figure 4C***), but not to the VA1lm PNs that are innervated by Or47b ORNs, rescued the reduction in synapse number and density (***Figure 4D,F***). When Ten-a was restored to Or47b-positive ORNs, defects in glomerular morphology persisted despite rescue of synaptic phenotypes (compare ***Figure 4C–A***), as partner matching occurs earlier than synaptogenesis (and before *Or47b-GAL4* turns on). Further consistent with a presynaptic role for Ten-a, knockdown of Ten-a in ORNs by RNAi also reduced the number and density of synapses similarly to the mutant (***Figure 4D,F***). Here, glomerular morphology was unaffected as VA1lm is already a 'Ten-a low' glomerulus and does not directly require Ten-a for partner matching. Interestingly, knockdown of *ten-m* in the Or47b ORNs did not result in any synaptic phenotype (***Figure 4D–F***), demonstrating specificity for *ten-a*. Thus, Ten-a is required presynaptically for proper synapse number and density, likely in a partially redundant fashion with additional molecules.

## Ultrastructural analysis of *ten-a* mutant synapses in the antennal lobe

Previous data and our Brp-Short analyses suggest that the assay is largely reflective of actual synapse number in neurons. However, we sought to confirm these results using a method independent of confocal microscopy or the Brp-Short label. We compared synapse number in ultrathin EM sections of wild-type and *ten-a* mutant flies (***Figure 5A–B***). Quantification of all synapses in a glomerulus (and thus

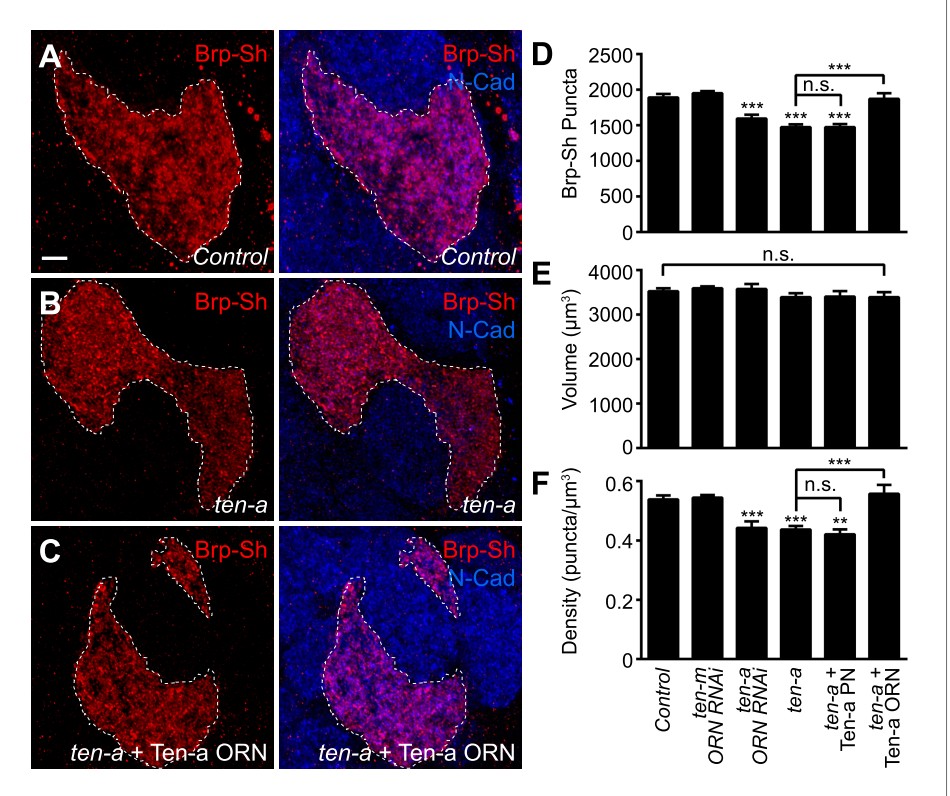

**Figure 4**. Presynaptic Ten-a is required for proper ORN synapse number and density. Representative high-magnification confocal stacks of Brp-Short puncta in VA1lm ORNs stained with antibodies against Brp-Short-mStraw (red) and N-Cadherin (blue) in control (**A**), *ten-a* null mutants (**B**) and *ten-a* null mutants where a Ten-a transgene is expressed in this specific class of ORNs using *Or47b-GAL4* (**C**). (**D**) Quantification of Brp-Short puncta in the above conditions as well as following presynaptic RNAi against *ten-a*, presynaptic RNAi against *ten-m*, or in *ten-a* null mutants where a Ten-a transgene is expressed in 2/3 of the PNs, including those innervated by Or47b-positive ORNs. (**E**) Quantification of ORN neurite volume in the same genotypes. n.s. = not significant. (**F**) Quantification of ORN synaptic density in the same genotypes. Significance was assessed with a one-way ANOVA and corrected for multiple comparisons by a posthoc Tukey's multiple comparisons test. **$p < 0.01$, ***$p < 0.001$. Unless otherwise noted, significance is compared to control. In all cases, the *ten-a* mutant and *ten-a* ORN RNAi phenotypes are statistically indistinguishable and, in the *ten-a* mutant, presynaptic Ten-a expression rescues the observed phenotypes. The disrupted glomerular shape is due to partner matching errors in the *ten-a* mutant (**Hong et al., 2012**), which is not rescued due to the late onset of *Or47b-GAL4* relative to the partner matching process. Dashed lines denote the glomeruli as delineated by Brp-Short label. In all cases, data represent mean ± SEM and n ≥ 7 animals, 14 antennal lobes. Scale bar = 5 μm.

The following figure supplements are available for figure 4:

**Figure supplement 1**. Representative images for additional genetic manipulations from *Figure 4*.

**Figure supplement 2**. *ten-a* phenotypes in DL4, a glomerulus with no morphology defects.

a direct comparison to the Brp-Short assay) would require serial reconstruction of that entire glomerulus by EM. However, if the Brp-Short assay accurately reflects changes in synapse number following genetic perturbation, we reasoned that this should be evident in the change in the number of T-bar profiles in individual sections at the EM level. Indeed, quantification of T-bar profiles showed a 27% reduction in the number of T-bars (*Figure 5I*). This is consistent with the results from the Brp-Short assay (*Figure 4D*), which shows a 23% reduction. The slight difference may reflect an under-report of the phenotype by our confocal-level analysis or may reflect the fact that the EM analysis does not differentiate between ORN, PN, and LN presynapses while the Brp-Short assay does. The largely similar magnitude of synapse reduction in EM and confocal studies supports the validity of the Brp-Short assay, and further substantiates that Ten-a is required for proper synapse number in the CNS.

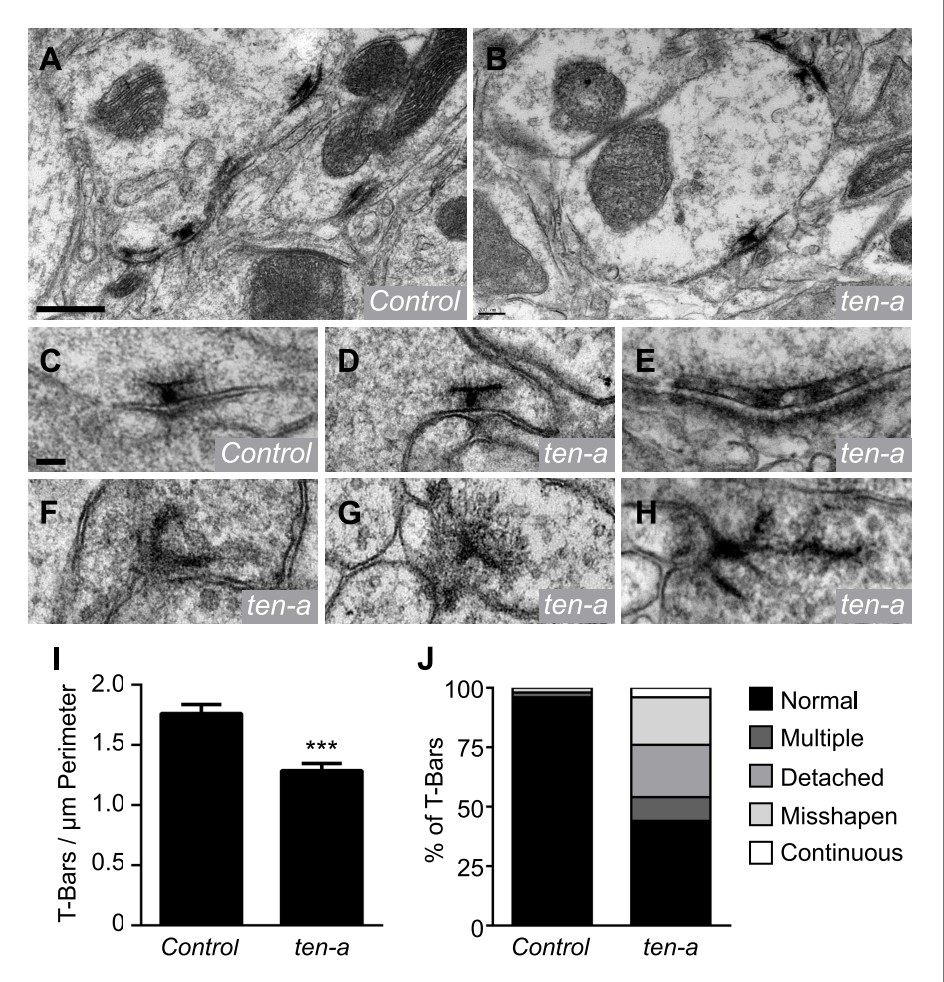

**Figure 5**. Ultrastructural analysis of active zones in *ten-a* mutants. Representative transmission electron microscopy (TEM) images of active zones in control (**A**) and *ten-a* null mutant antennal lobes (**B**). T-bar profiles are visible in both genotypes, though reduced in number in the *ten-a* mutant. Scale bar = 500 nm. (**C**) High magnification TEM of a single active zone in a control animal. Note the normal T-bar morphology. (**D–H**) High magnification TEM images of active zones in *ten-a* mutants. In some cases, normal (**D**) morphology is observed. In the majority of cases, defects including multiple T-bars (**E**), misshapen T-bars (**F**), detached T-bars (**G**), and continuous electron dense material (**H**) are evident. Scale bar = 100 nm. (**I**) Quantification of T-bars per unit perimeter of measured antennal lobe terminals. All terminals were measured, including ORNs, PNs, and LNs. *ten-a* mutants display a 27% reduction from control animals. Data represent mean ± SEM. (**J**) Distribution of T-bar defects as a percentage of the total T-bars. Here, *n* = 1103 T-bar per unit perimeter measurements from three animals for control and *n* = 847 T-bar per unit perimeter measurements from three animals for *ten-a*.

At neuromuscular synapses, Teneurins are required for proper T-bar structure. When Teneurins are absent, T-bars display defects consistent with failed synaptogenesis (***Mosca et al., 2012***). To determine whether Ten-a was also required in olfactory neurons for T-bar structure, we examined individual T-bar morphology in ultrathin sections of wild-type and *ten-a* antennal lobes. In wild-type animals, T-bar morphology was nearly invariant (***Figure 5C***), with infrequent occurrences of misshapen or multiple T-bars (***Figure 5J***). In *ten-a* mutants, 44% of T-bar profiles appeared normal (***Figure 5D,J***). The remaining 56% displayed notable structural defects including continuous, multiple T-bars (***Figure 5E***), misshapen T-bars (***Figure 5F***), and T-bars that were detached from the synaptic membrane (***Figure 5G***). These defects resembled those observed at *ten-a* mutant NMJs (***Mosca et al., 2012***). We also observed continuous, electron-dense material resembling undivided T-bars (***Figure 5H***) that was not found previously. Thus, Ten-a is required not just for the normal number of active zones, but also for their proper

structure in ways both consistent with and beyond that of previous studies. Specifically, *ten-a* may be required for the maturation of synapses, as misshapen and undivided T-bars resemble immature synapses (*Owald et al., 2010*).

## Presynaptic spectrin is regulated by ten-a and required for normal synapse number

Teneurins can regulate cytoskeletal architecture (*Nunes et al., 2005*; *Morck et al., 2010*; *Suzuki et al., 2014*) and function at the *Drosophila* NMJ by organizing spectrin, a cytoskeletal component (*Mosca et al., 2012*). However, it is unknown whether this function of the Teneurins is conserved in the CNS. To further probe the mechanism by which Teneurins regulate central synapse number and to determine whether cytoskeletal regulation is involved, we compared levels of spectrin staining in the antennal lobe between wild-type and *ten-a* mutant flies. We found that spectrin immunofluorescence in the antennal lobe was reduced in the *ten-a* mutant compared to wild-type (*Figure 6A–B*) while N-Cadherin, a marker of synaptic neuropil, was unaffected (*Figure 6F*). This was observed for α- and β-spectrin (*Figure 6F*, *Figure 6—figure supplement 1*, panels A–D), which function as a heterotetramer (*Machnicka et al., 2014*).

To test whether proper spectrin level is required for synapse number, we used ORN-specific knockdown of either α- or β-spectrin and measured Brp-Short puncta. In both cases (*Figure 6D–E*), RNAi in Or47b-positive ORNs resulted in a 17–18% reduction in Brp-Short puncta (*Figure 6G*). This effect, though slightly weaker, phenocopied the reduction seen in the *ten-a* mutant. We observed a similar effect in another ORN class, the Or67d-positive neurons (*Figure 6—figure supplement 1*), further supporting a role for spectrin in regulating olfactory synapse number. Moreover, the *ten-a* phenotype was not enhanced by concurrent RNAi knockdown of α-spectrin in ORNs (*Figure 6G*), suggesting that Ten-a and α-spectrin function in the same pathway to regulate olfactory synapse number.

## Labeling postsynaptic acetylcholine receptor clusters via the Dα7 subunit

A functional synapse consists of a presynaptic neurotransmitter release site and a postsynaptic neurotransmitter receptor cluster (*Figure 1A*). Therefore, critical parameters of synaptic organization within a circuit not only include the location and number of presynaptic active zones, but also postsynaptic receptor clusters. Therefore, we also examined the organization of postsynapses. Given that ORNs are cholinergic (*Wilson, 2013*), an ideal labeling strategy would image postsynaptic acetylcholine receptors.

We selected the Dα7 acetylcholine receptor subunit because it is endogenously expressed in the antennal lobe (*Fayyazuddin et al., 2006*) and it has been used to examine organization in the mushroom body, a higher olfactory center (*Leiss et al., 2009a*, *2009b*; *Kremer et al., 2010*; *Christiansen et al., 2011*). We used a GFP-tagged Dα7 transgene (*Leiss et al., 2009b*) under the control of the GAL4/UAS system (*Figure 7A*) to visualize postsynapses in vivo. Expression of Dα7-GFP in PNs revealed distinct puncta, possibly corresponding to acetylcholine receptor (AChR) clusters (*Figure 7B*). These puncta were apposed to endogenous Brp puncta, as revealed by nc82 staining (Supplement 1 to *Figure 7*), consistent with these puncta representing *bona fide* synapses. To examine AChR clusters in PNs, we co-expressed Dα7-GFP with mtdT as a general neurite label (*Figure 7B*). As such, the approach is analogous to our Brp-Short assay and yielded similar results, enabling a quantitative assessment of the number (*Figure 7C*) and density (*Figure 7D*) of AChR clusters.

As we are limited to genetically accessible PN subsets, we focused on identifying organizational parameters in the PNs that innervate the DA1 and VA1d glomeruli via the *Mz19-GAL4* driver (*Ito et al., 1998*). As with ORN presynapses, the assay revealed that the number of AChR puncta scales with glomerular size (*Figure 7E*). Further, known sex-specific differences in DA1, as seen in glomerular volume (*Stockinger et al., 2005*) and in ORN synapses (*Figure 1*), were also observed (*Figure 7E*). The differences between the Brp-Short and AChR assays for the DA1 (1632 ± 22 for Brp-Short vs 2007 ± 48 for Dα7) and VA1d (1107 ± 18 for Brp-Short vs 1762 ± 38 for Dα7) glomeruli may reflect the fact that the Brp-Short assay does not distinguish ORN synapses onto PNs and LNs, and the Dα7-GFP assay does not distinguish synapses from ORNs and LNs onto PNs. As these values are less than twofold different, this is consistent with the majority of synapses labeled being ORN to PN synapses (*Kazama and Wilson, 2009*; *Wilson, 2013*). The similarity between the numbers of endogenous Brp and Brp-Short puncta suggests that Brp-Short is a more accurate estimator of absolute synapse number. The larger number of AChRs detected in each glomerulus may reflect an overestimation associated with full-length Dα7 overexpression or that these are postsynaptic not just to cholinergic ORNs,

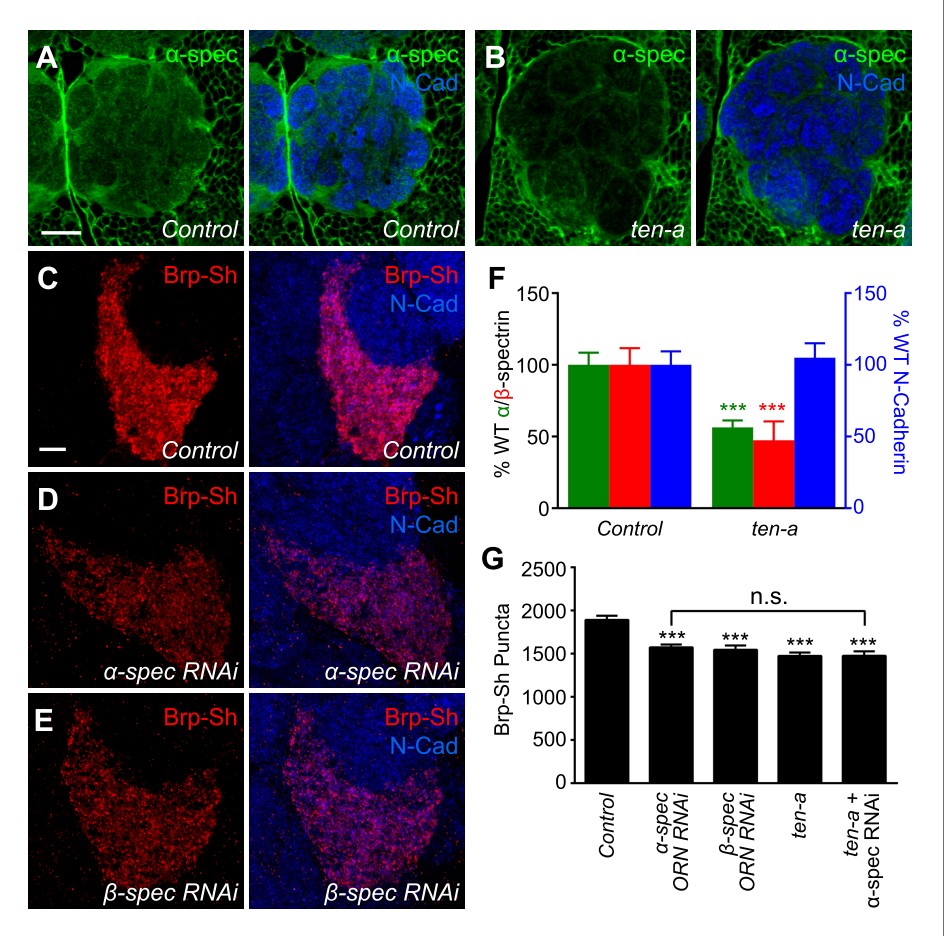

**Figure 6**. Ten-a and spectrin function together for normal synapse number. (**A–B**) Representative single confocal optical sections taken at equivalent positions of antennal lobes stained for α-spectrin and N-Cadherin in control (**A**) and *ten-a* mutant (**B**) adults. In *ten-a* mutants, α-spectrin staining is reduced compared to control. The disrupted morphology of the antennal lobe itself is due to partner matching errors evident in the *ten-a* mutant (*Hong et al., 2012*). Scale bar = 20 µm. (**C–E**) Representative confocal Z-stack images of the ORNs in the VA1lm glomerulus expressing Brp-Short in control animals (**C**), or animals expressing dsRNA against α-spectrin (**D**) or β-spectrin (**E**), and stained with antibodies against Brp-Short (red) and N-Cadherin (blue). (**F**) Quantification of α-spectrin (green), β-spectrin (red), and N-Cadherin (blue) immunofluorescence in control and *ten-a* mutants. ***p < 0.001. (**G**) Quantification of Brp-Short puncta in the noted genotypes. In all cases, similar reductions in puncta number are observed. Moreover, the genetic perturbations do not enhance each other, suggesting function in the same pathway. Significance was assessed with a one-way ANOVA and corrected for multiple comparisons by a posthoc Tukey's multiple comparisons test. ***p < 0.001 (compared with control). In all cases, data represent mean ± SEM and n ≥ 10 animals, 19 antennal lobes. Scale bar = 5 µm.

The following figure supplement is available for figure 6:

**Figure supplement 1**. *ten-a* regulates spectrin levels and spectrin regulates synapse number.

but also other excitatory neurons such as local interneurons (*Chou et al., 2010*) or PN-PN chemical synapses (*Ng et al., 2002*; *Wilson et al., 2004*).

Calculation of AChR puncta density in PNs revealed subtle but significant differences across different glomeruli. In the VA1d glomerulus, the densities were identical between males and females (*Figure 7F*). However, these were different from AChR puncta densities in the DA1 glomerulus. There was a modest but significant difference between both male and female AChR densities in DA1 and between both DA1 AChR densities and the shared VA1d AChR density (*Figure 7F*). Unlike ORNs, where the Brp-Short density was identical across different classes of neurons, PNs can have different

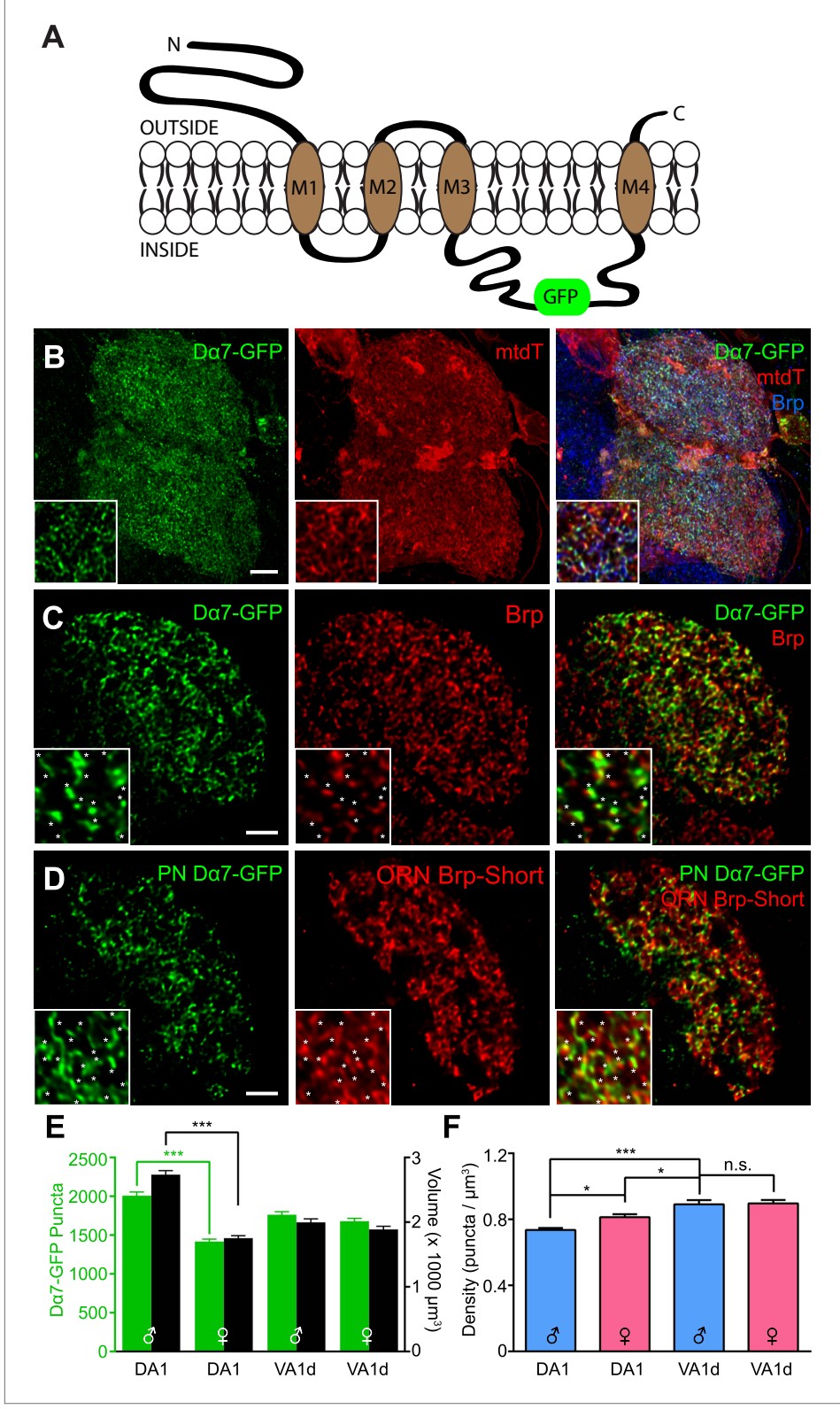

**Figure 7**. Measuring postsynaptic acetylcholine receptor clusters with Dα7-GFP. (**A**) Diagram of the GFP-tagged Dα7 acetylcholine receptor subunit used for AChR visualization. (**B**) High magnification confocal z-stack images of PNs in the DA1 and VA1d glomeruli expressing Dα7-GFP and mtdT, stained with antibodies against GFP (green), mtdT (red), and endogenous Brp (blue). Patches of mtdT labeling represent ascending PN axons within the plane

*Figure 7. Continued on next page*

*Figure 7. Continued*

of the glomerulus. Insets show a high magnification single optical section demonstrating the punctate nature of Dα7-GFP. (**C**) Representative high magnification single optical sections of Mz19-positive PNs in the DA1 glomerulus expressing Dα7-GFP and stained with antibodies against GFP (green) and endogenous Brp (red). The majority of GFP-positive puncta are apposed to or colocalized with endogenous Brp (yellow), consistent with their association with *bona fide* active zones. Insets show high magnification of a single optical section where asterisks denote apposed Dα7 puncta. Brp puncta without apposition likely belong to synapses not labeled by the PN-GAL4 driver. Scale bar = 5 μm (2 μm for inset). (**D**) Representative high magnification single optical sections of Mz19-positive PNs and Or67d-positive ORNs in the DA1 glomerulus expressing Dα7-GFP in the PNs and Brp-Short-mStraw in the ORNs using two binary expression systems. Most ORN active zones (labeled by Brp-Short, red) are apposed to or colocalized (yellow) with PN Dα7 puncta (green), further supporting that these puncta label *bona fide* synaptic contacts. Insets show high magnification of a single optical section where asterisks denote apposed Brp-Short::Dα7 pairs. Brp puncta without apposition likely belong to ORN synapses with neurons other than PNs and Dα7 puncta without apposition likely correspond to PN postsynaptic sites apposed to synaptic contacts from neurons other than ORNs. (**E**) Quantification of Dα7 AChR puncta (green, left axis) and neurite volume (black, right axis) in the DA1 and VA1d glomeruli of both male and female adult flies. Statistical comparisons between males and females of a single genotype were done by student's *t* test. ***p < 0.001. (**F**) Quantification of AChR density in male (blue) and female (pink) adults based on the data from (**C**). Significance was assessed with a one-way ANOVA and corrected for multiple comparisons by a posthoc Tukey's multiple comparisons test. *p < 0.05. n.s. = not significant. In all cases, data represent mean ± SEM and *n* ≥ 11 animals, 21 antennal lobes. Scale bar = 5 μm.

densities between distinct glomeruli and even between sexes for the same glomerulus. Thus, the parameters that govern presynaptic density may differ from those that govern postsynaptic density in the same glomerulus.

## Teneurins are required for proper AChR number in antennal lobe PNs

To further examine if the Teneurins regulate postsynaptic acetylcholine receptor number and density, we utilized the Dα7-GFP assay to determine the effect of Teneurin perturbation on AChR puncta number. We examined PNs in DA1 and VA1d (*Figure 8A*) and counted the AChR puncta of both glomeruli together as one measurement, as partner matching defects following Teneurin perturbation make it difficult to differentiate between the two glomeruli (*Hong et al., 2012*). In *ten-a* mutants (*Figure 8B*), the number of AChR clusters in these glomeruli was decreased by 23%, compared to wild type (*Figure 8D*). This is consistent with results from the Brp-Short assay. Moreover, PN neurite volume was unaffected (*Figure 8E*), so AChR puncta density was similarly reduced (*Figure 8F*). Thus, two independent assays, both pre- and postsynaptic, show the same phenotypes, demonstrating a clear effect of *ten-a* loss on synapse organization in olfactory neurons.

At the NMJ, presynaptic Ten-a functions largely in a transsynaptic, heterophilic complex with postsynaptic Ten-m to regulate synapse organization (*Mosca et al., 2012*). Ten-a functions presynaptically in ORNs to ensure proper synapse number (*Figure 4*). Thus, we hypothesized that the loss of Ten-m in postsynaptic PNs should result in a similar phenotype. As the *ten-m* mutant is larval lethal (*Zheng et al., 2011*), we expressed a previously validated transgenic RNAi line against *ten-m* (*Hong et al., 2012*; *Mosca et al., 2012*) in Mz19 PNs and quantified AChR puncta number using the Dα7 assay (*Figure 8C*). *ten-m* knockdown phenocopied the *ten-a* phenotype (*Figure 8D*). As above, PN neurite volume was unaffected (*Figure 8E*), leading to a concomitant decrease in AChR puncta density that also phenocopied the *ten-a* phenotype (*Figure 8F*). Further, this reduction was not enhanced by knocking down *ten-m* in PNs of a *ten-a* null mutant (*Figure 8D–F*), suggesting that the two function in the same genetic pathway.

## Ten-m regulates presynaptic active zone number transsynaptically

Though genetic evidence suggests that Ten-m functions with Ten-a transsynaptically to regulate synapse number, it could conceivably regulate AChR number independently of how Ten-a regulates active zone number. Thus, we sought to more explicitly test a transsynaptic role of Ten-m in regulating proper active zone number. Due to the availability of genetic access to both the presynaptic ORNs and the postsynaptic PNs, we examined the DA1 glomerulus. Further, DA1 is a *ten-m* low glomerulus (*Hong et al., 2012*), allowing us to specifically test the role of the basal level of Ten-m. Using the GAL4 and QF systems, we simultaneously labeled Or67d-positive ORNs with QUAS Brp-Short-mStraw using *Or67d-QF*

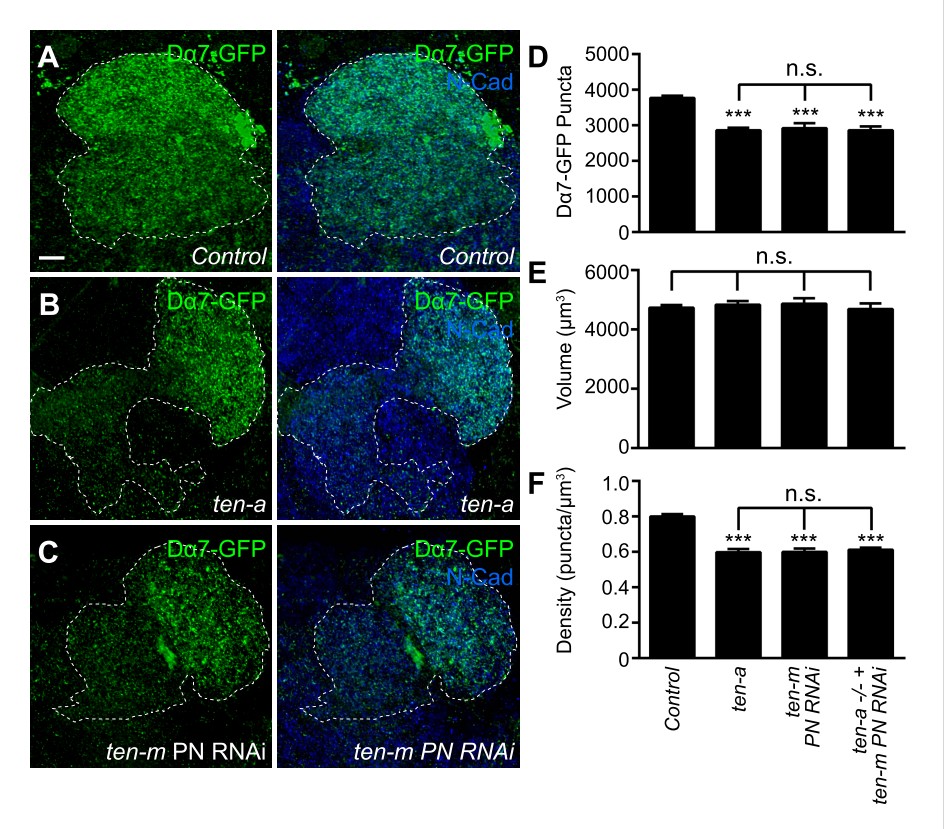

**Figure 8**. Teneurins function to regulate acetylcholine receptor number. (**A**–**C**) Representative high magnification confocal z-stack images of DA1 and VA1d PNs expressing Dα7-GFP in control animals (**A**), *ten-a* mutants (**B**), or animals expressing RNAi against *ten-m* in the same PNs (**C**), and stained with antibodies against GFP (green) and N-Cadherin (blue). DA1 and VA1d glomeruli are outlined together (white dashed line) and were determined using the accompanying mtdT label (not shown). Due to partner matching defects in the *ten-a* mutant that prevent clear delineation between the two glomeruli, the combined count was compared across genotypes. (**D**) Quantification of Dα7-GFP puncta in the noted genotypes. (**E**) Quantification of PN neurite volume in all genotypes. (**F**) Quantification of AChR density in all genotypes. All observed phenotypes are statistically indistinguishable. Significance was assessed with a one-way ANOVA and corrected for multiple comparisons by a posthoc Tukey's multiple comparisons test. ***p < 0.001 in comparison with control. n.s. = not significant. In all cases, data represent mean ± SEM and $n \geq 8$ animals, 16 antennal lobes. Scale bar = 5 μm.

(*Liang et al., 2013*) and expressed transgenic RNAi against *ten-m* in DA1 PNs using *Mz19-GAL4* (*Figure 9A*). Control animals displayed a similar number of Brp-Short puncta as the UAS construct (*Figure 9B,D*), demonstrating the validity of this reagent. Postsynaptic *ten-m* knockdown, however, reduced the number of presynaptic Brp-Short puncta by 20% (*Figure 9C,D*). This reduction is similar to the proportion by which *ten-m* RNAi reduced AChR number (*Figure 8D*) and also to the proportion by which loss of *ten-a* reduced Brp-Short number (*Figure 4D*). This suggests that Ten-m is the postsynaptic partner of Ten-a in regulating presynaptic active zone number, and that the presynaptic Ten-a/ postsynaptic Ten-m interaction that regulates NMJ synapses is conserved in olfactory synapses.

## Discussion

Understanding the development and organization of synapses between identified neurons within complex circuits in the brain represents a major goal of neuroscience. To address these questions, techniques are needed to visualize synapses at the light microscopy level in an intact system that is amenable to genetic manipulation. Here, we identify parameters that govern synapse number, density, and subcellular organization using two fluorescently-tagged synaptic proteins expressed from single transgenes in combination with high-resolution confocal microscopy and image processing to visualize

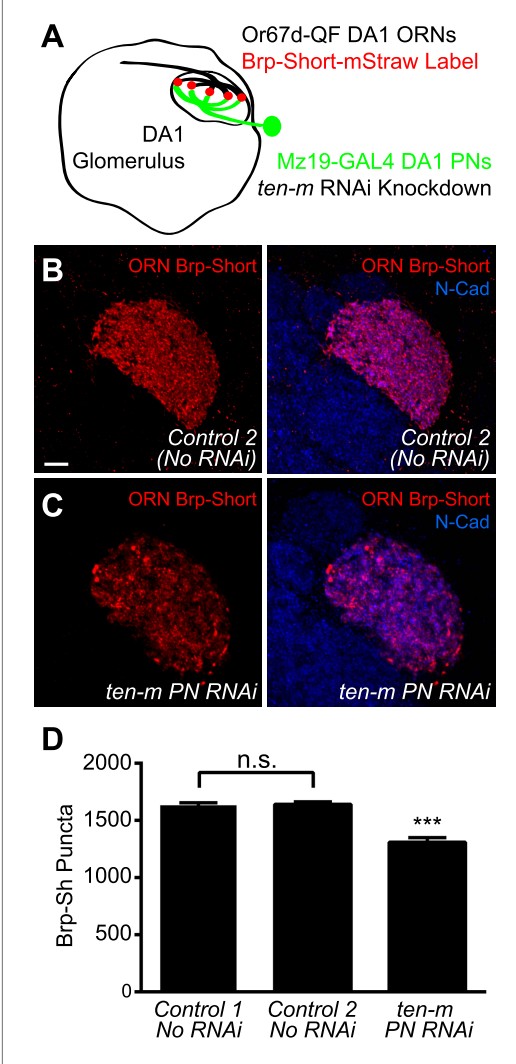

**Figure 9**. Ten-m regulates presynaptic active zone number transsynaptically. (**A**) Diagram of the antennal lobe with the DA1 glomerulus outlined showing the experimental design. Or67d-positive ORN axon terminals that innervate DA1 are outlined in black; their active zones are labeled by *Or67d-QF*-driven *QUAS-Brp-Short-mStraw*. *Mz19-GAL4*-positive PNs that project dendrites to DA1 express UAS-RNAi against *ten-m* in experimental animals. (**B–C**) Representative high magnification confocal z-stack images of DA1 ORNs expressing Brp-Short in control animals (**B**) or in animals concurrently expressing *ten-m* RNAi in the DA1 and VA1d PNs (**C**) and stained with antibodies against mStraw (red) and N-Cadherin (blue). (**D**) Quantification of Brp-Short puncta in the noted genotypes. Controls 1 and 2 represent Brp-Sh puncta assayed by GAL4/UAS (as in **Figure 1**) and QF/QUAS binary system binary system, respectively, and are not significantly different, as assayed by student's *t* test (n.s.). Significance of *ten-m* PN RNAi was assayed by student's *t* test between it and Control 2. ***p < 0.001. In all cases, data represent mean ± SEM and *n* ≥ 5 animals, 10 antennal lobes. Scale bar = 5 μm.

synapses in the *Drosophila* olfactory system in vivo. We demonstrate that these methods are amenable to analysis in both individual neuronal classes and individual neurons. Our study provides a synapse-level analysis of innervation of olfactory receptor neurons, projection neurons, and local interneurons in the antennal lobe, which has emerged as a model circuit for analyzing principles of information processing (*Wang, 2012*; *Wilson, 2013*; *Twick et al., 2014*). Finally, using these synaptic tagging assays, we show that the Teneurins are required for the proper synapse number in ORNs and PNs as well as the structure of the active zones themselves. This reveals a critical role for these transmembrane proteins in organizing central synapses, likely by regulating the cytoskeleton. These approaches can be broadly used to study synaptic organization of neurons for which genetic access is available, and to investigate the functions of any other proteins in the organization and development of CNS synapses.

## Quantitative synaptic tagging assays to study central synapses

In *Drosophila*, previous approaches to studying central synapses used tagged synaptic vesicle proteins (*Ito et al., 1998*; *Robinson et al., 2002*) to reveal putative synaptic sites. While consistent with synapses, they could also stain non-synaptic, trafficking vesicles. We utilized a structural active zone component, Brp (*Wagh et al., 2006*). By expressing a truncated Brp transgene, Brp-Short (*Schmid et al., 2008*), using the GAL4/UAS system, this approach can label synapses in any neurons with genetic access. Brp-Short expression requires endogenous Brp for proper localization (*Figure 1— figure supplement 1*) and does not alter function or morphology (*Fouquet et al., 2009*; *Kremer et al., 2010*). Thus, it accurately reports endogenous active zones. Recently, an elegant technique, STaR, was developed, which tags an additional, BAC-sourced copy of Brp with an epitope tag whose expression is conditional upon FLP-recombinase-mediated excision of an intervening stop codon (*Chen et al., 2014*). An important advantage of STaR is that Brp expression is controlled by its endogenous promoter, thus guarding against mislocalization of Brp or perturbation of synaptic function due to overexpression. A caveat of Brp-Short is that it is controlled by GAL4 and thus the levels may be different from the endogenous level. While Brp-Short overexpression does not interfere with synaptic function, care must be taken not to overexpress it to a level that saturates the active

zone localization machinery when utilized in new cell types. The advantages of Brp-Short over STaR are (a) that it does not require a cell type-specific FLP transgene, which is not as widely available as GAL4 lines, or a BAC-bearing copy of Brp, and (b) that it can be co-expressed with UAS-transgenes for rescue or RNAi for perturbation experiments, although STaR can also achieve this aspect by using a cell-type-specific GAL4 and an extra *UAS-FLP* transgene.

To examine putative postsynaptic acetylcholine receptor clusters, we use a GFP-tagged subunit, Dα7, to study cholinergic synapses in the antennal lobe. This transgene has been used to examine synaptic organization (*Leiss et al., 2009a*, *2009b*; *Kremer et al., 2010*; *Christiansen et al., 2011*). Though false positives can be associated with full-length protein overexpression, our observation of endogenous Brp puncta apposed to these Dα7-GFP puncta suggests that these receptors are properly localized to endogenous synapses. This assay complements the Brp-Short presynaptic assay and can be adapted for other tagged postsynaptic receptor transgenes.

## Synaptic organization in the Drosophila antennal lobe

Though considerable advances have been made in our understanding of the wiring specificity (*Hong and Luo, 2014*) and physiological properties (*Wilson, 2013*) of the *Drosophila* olfactory circuit in the antennal lobe, little is known about its synaptic organization. As this system has emerged as a model neural network, a detailed mapping of synaptic organizational principles is integral towards advancing the study of circuit dynamics. Indeed, the juxtaposition of distinct types of synapses between multiple neuronal classes in the antennal lobe provides a model to study complex synaptic interactions compared to a neuromuscular synapse that features only two synaptic partners: the motoneuron and the muscle. We utilized the Brp-Short and Dα7 assays to probe how synapses in ORNs, PNs, and LNs are organized in the antennal lobe with respect to their number, density, and location. This work offers the first comprehensive information on (1) the number of active zones made by each ORN within a glomerulus, (2) the stereotypy of synapse numbers between those individual ORNs, (3) the prominence of PN presynaptic inputs within a glomerulus, suggesting a robust feedback mechanism, and (4) the relatively small contribution of LN active zones to the antennal lobe circuit. Our analyses suggest distinct rules that govern the synaptic organization of antennal lobe neurons. ORNs, the primary input neurons of the olfactory system, are diverse in their olfactory receptor expression, ligand specificity, and glomerular targeting specificity (*Liang and Luo, 2010*); we now show that they also differ in the absolute synapse number (*Figure 1*). However, despite such differences, all five classes we examined (DA1, VA1d, VA1lm, DL4, and DM6 ORNs) have identical synaptic densities, suggesting that this represents a general rule for other ORN classes. This may further suggest that the primary job of the ORN is to convey information from the environment into the system as faithfully as possible, that all information is treated equally at this level, and that weighting computations occur downstream in the brain (*Masse et al., 2009*; *Wilson, 2013*). Indeed, our analyses indicate that each ORN makes an equivalent, discrete number of synapses within a given glomerulus with little variation (*Figure 2*), further supporting this hypothesis.

Interestingly, the density of postsynaptic receptors differs between glomeruli and even within the same glomerulus between sexes (*Figure 7*). This variation can be due to technical caveats, such as *Mz19-GAL4* does not label all PNs that innervate DA1 and VA1d (*Jefferis et al., 2001*; *Liang et al., 2013*), or that Dα7-GFP clusters do not reflect the absolute number of AChRs. As the relative numbers still show these differences, an interesting possibility suggested by these results is that postsynaptic PN AChRs already reflect a transformed olfactory representation compared to output synapses of ORNs. The difference in PN AChR density as compared to the constant density of ORN active zones suggests that different classes of PNs may modulate how information is received by regulating the number of acetylcholine receptors. This can thus contribute to the transformation of olfactory representation by antennal lobe neurons (*Wilson, 2013*).

Fine-scale analysis of synapse localization within projections of ORNs, LNs, and PNs (*Figure 3*) suggested that these three types of neurons differ in their subglomerular organization. While occupying the vast majority of territory throughout the entire glomerulus, ORN processes and synapses leave distinct voids. A significant proportion of LNs form synapses in these voids, likely to other LNs or PNs. However, there is also overlap between LN and ORN processes and synapses, consistent with physiologically characterized ORN → LN and LN → ORN synapses (*Olsen et al., 2007*, *2010*; *Root et al., 2007*; *Kazama and Wilson, 2008*, *2009*; *Olsen and Wilson, 2008*; *Root et al., 2008*; *Huang et al., 2010*; *Yaksi and Wilson, 2010*; *Root et al., 2011*; *Wilson, 2011*, *2013*). Within their respective neurites,

ORNs and PNs display uneven distributions and synaptic clusters to varying degrees while active zones in LNs are more evenly distributed throughout their processes. These characterizations contribute to the growing repertoire of studies seeking to understand synaptic organization in *Drosophila* olfactory circuits (*Murthy et al., 2008*; *Caron et al., 2013*; *Fisek and Wilson, 2014*), adding synapse-level imaging to physiological techniques. Finally, the existence of synaptic organization parameters detailing number and location suggests molecular mechanisms designed to enforce those rules, both at cellular and circuit levels. Indeed, such analysis represents an integral part of neuronal circuit analysis, as recent work on retinal neurons has shown that accounting for synapse position is a critical aspect of modeling connectivity (*Kim et al., 2014*).

## Central synapse density is regulated by a conserved Teneurin mechanism

Our data demonstrate that the Teneurins, a family of transsynaptic adhesion molecules, regulate one of these synaptic organizational paradigms: synapse number. Beyond molecules like RPTPs (*Takahashi and Craig, 2013*) and Wnts (*Park and Shen, 2012*), there is little known conservation between the organization mechanisms of central synapses and the NMJ. In vertebrates, previous studies have identified a number of synaptogenic signaling and cell adhesion molecules in the CNS (*Gerrow and El-Husseini, 2006*; *Missler et al., 2012*), but in many cases, their roles at the NMJ are either minimal or unknown. Likewise, the roles of pathways including Rapsyn, Dok7, MuSK, and Tid1, which are well established at the NMJ (*Wu et al., 2010*), have no well-established roles at CNS synapses. Among the identified central synaptogenic molecules, no master controller of synapses, like Agrin, has been discovered (*Shen and Scheiffele, 2010*). In the mammalian CNS, synaptic adhesion molecules like Neurexin and Neuroligin have demonstrated organizational roles (*Sudhof, 2008*) but their role (if any) at the NMJ is largely unknown. In *Drosophila*, considerable work has been done at the NMJ to understand synapse formation and organization (*Menon et al., 2013*). For example, neuromuscular Neurexin and Neuroligin regulate synaptic development and assembly (*Li et al., 2007*; *Banovic et al., 2010*; *Sun et al., 2011*; *Chen et al., 2012*; *Mosca et al., 2012*; *Owald et al., 2012*), but remain (as with many other identified molecules) largely untested in the CNS due to the absence of techniques for doing so.

Our approaches have now enabled such an examination, uncovering a strongly conserved synaptic organization function of the Teneurins from PNS to CNS. Recent work has shown that these evolutionarily conserved proteins are involved in synaptic partner matching between neurons in the *Drosophila* olfactory system and between muscles and motoneurons at the NMJ (*Hong et al., 2012*; *Mosca et al., 2012*). As there are marked differences between central and peripheral synapses (like the NMJ), it is further unclear whether the mechanisms would be conserved or if they would be wholly different. Here we find, as assayed by Brp-Short (*Figure 4*), ultrastructural (*Figure 5*), and Dα7 (*Figure 8*) analyses, that Teneurins in olfactory neurons are required for normal synapse number. Teneurin perturbation also reduces synaptic density, a parameter that is highly invariable for ORNs under normal conditions (*Figure 1I*). We determined that presynaptic Ten-a in ORNs (*Figure 4*) likely functions with postsynaptic Ten-m in PNs (*Figures 8,9*) to regulate levels of the spectrin cytoskeleton (*Figure 6*). Our evidence is consistent with spectrin and the Teneurins functioning in the same genetic pathway to regulate synapse organization and density. The perturbation of active zone number in ORNs by knocking down *ten-m* in PNs further suggests that Teneurins regulate synapse number and organization in the CNS via a transsynaptic mechanism. This highlights further conservation between central and peripheral synapse organization in the use of Teneurins. There are, however, some differences between the CNS and the PNS regarding the Teneurins. While presynaptic *ten-m* has a minor role in synaptic organization at the NMJ (*Mosca et al., 2012*), our data suggests the lack of such a role in the CNS (*Figure 4D*). Thus, these different systems may also use the Teneurins differently. Mammalian Teneurins organize the visual system (*Leamey et al., 2007*; *Suzuki et al., 2012*; *Antinucci et al., 2013*; *Young et al., 2013*) and Ten-2 can serve as a ligand for Latrophilin and localize to synapses in cultured neurons (*Silva et al., 2011*; *Boucard et al., 2014*). However, as proper synaptic function is impaired in many neuropsychiatric disorders (*Guilmatre et al., 2014*) and human Teneurin-4 is associated with increased susceptibility to bipolar disorder (*Psychiatric GWAS Consortium Bipolar Disorder Working Group, 2011*), understanding how the Teneurins regulate central synapses is a question with clinical relevance. Studies in vivo will be important to determine how mammalian Teneurins regulate synaptic organization and whether the different Teneurins can have specific roles at synapses.

In summary, our results demonstrate a role for the Teneurins in regulating the number of central synapses and highlight mechanistic conservation between peripheral and central synapse formation. Moreover, the fact that ORNs can be mistargeted but still have the correct number of synapses suggests that target choice and synapse organization can be biologically separable, even when they employ the same molecules.

## Materials and methods

### Drosophila genetics and transgenic lines

The following GAL4 driver lines were used to restrict expression to individual classes of neurons: *AM29-GAL4* (DL4 and DM6 ORNs; *Endo et al., 2007*), *Or47b-GAL4* (VA1lm ORNs; *Vosshall et al., 2000*), *Or67d-GAL4* (DA1 ORNs; *Kurtovic et al., 2007*), *Or88a-GAL4* (VA1d ORNs; *Vosshall et al., 2000*), *Mz19-GAL4* (DA1, DC3, and VA1d PNs; *Jefferis et al., 2004*), *Pebbled-GAL4* (all ORNs; *Sweeney et al., 2007*), *LN5(NP3056)-GAL4* (~50 pan-glomerular LNs; *Chou et al., 2010*). The *GH146-QF* line (*Potter et al., 2010*) was used to genetically access 2/3 of all PNs. The following transgenes were used for labeling or genetic manipulation: *UAS-Brp-Short-mStraw* (*Fouquet et al., 2009*), *UAS-Brp-Short-EGFP* (*Schmid et al., 2008*), *UAS-Dα7-GFP* (*Leiss et al., 2009b*), *UAS-3xHA-mtdT* (*Potter et al., 2010*), *UAS-mCD8-GFP* (*Lee and Luo, 1999*), *UAS-DSyd1-EGFP* (*Owald et al., 2010*), *UAS-α-spectrin-dsRNA* (*Pielage et al., 2005*), *UAS-β-spectrin-dsRNA* (*Pielage et al., 2005*), *UAS-Ten-a* (*Mosca et al., 2012*), *UAS-Brp-IR-104422* (Vienna *Drosophila* RNAi Center), *UAS-Dcr2* (*Dietzl et al., 2007*), *UAS-ten-a-RNAi-V32482*, *UAS-ten-m-RNAi-V51173* (*Hong et al., 2012*; *Mosca et al., 2012*), *QUAS-mCD8-GFP* (*Potter et al., 2010*), *QUAS-Ten-a* (*Hong et al., 2012*), and *QUAS-Brp-Short-mStraw* (see below). The *Df (X) ten-a* line was used as a *ten-a* null mutant (*Hong et al., 2012*).

### QUAS-Brp-Short-mStrawberry transgenic flies

Brp-D3-mStraw (Brp-Short-mStraw) was amplified by PCR from the pTWmStraw-Brp-D3 vector (*Owald et al., 2010*; a kind gift of Stephan Sigrist) using the forward primer 5′-CACCATGGGAACTAGTGAC-3′ and the reverse primer 5′-CTAGCTTACGTCACG-3′ with a CACC appended to the 5′ end of the forward primer for subcloning into the Gateway (Invitrogen, Carlsbad, CA) pENTR/D-TOPO vector. Following a BP reaction to produce pENTR-Brp-Short-mStraw, this plasmid was recombined into the destination vector pQUAST-attB-Gateway (*Mosca et al., 2012*; a kind gift of Vincenzo Favaloro) using LR clonase. The resulting pQUAST-attB-Brp-Short-mStraw vector was transformed into the ΦC31 landing site 86Fb on the third chromosome using standard methods.

### Mosaic analyses

hsFLP MARCM analyses were conducted as described (*Lee and Luo, 1999*; *Joo et al., 2013*) with the following modification: to obtain small DL4 and DM6 ORN clones, animals were heat-shocked between 48 hr and 72 hr after egg-laying at 37°C for 20 min.

### Immunofluorescence and antibodies

Adult brains were dissected at 10 days post-eclosion and fixed, stained, and processed as previously described (*Wu and Luo, 2006*). The following antibodies were used: rat anti-N-Cadherin [DN-EX#8; 1:40, Developmental Studies Hybridoma Bank (DSHB)], mouse anti-Brp (NC82; 1:40, DSHB), rabbit anti-dsRed (Living Colors DsRed Polyclonal Antibody, 1:250, Clontech), chicken anti-GFP (GFP-1020; 1:1000, Aves Lab), mouse anti-α-spectrin (3A9; 1:50, DSHB), rabbit anti-β-spectrin (1:500, gift of R Dubreuil). Alexa488-, Alexa568-, and Alexa647-conjugated secondary antibodies were used at a concentration of 1:250 (Invitrogen, Carlsbad, CA). Samples were mounted in SlowFade Gold Antifade medium (Life Technologies, Grand Island, NY) by bridge mount under #1.5 cover slips.

### Imaging, analysis, and quantification for synaptic labels

Images were obtained on a Zeiss LSM510 Meta confocal laser-scanning microscope (Carl Zeiss, Oberkochen, Germany) using a 63× 1.4 NA PlanApo lens at an optical zoom of 3×. Images were centered on the glomerulus in question and the Z-boundaries set according to the confines of the synaptic label (Brp-Short-mStraw or Dα7-GFP). System gain was set to minimize saturation but maintain a high signal-to-noise ratio. Selected raw image stacks are available at http://web.stanford.edu/group/luolab/ALsynapse.shtml.

Raw image stacks for quantification were first deconvolved using AutoQuant X3 software (Media Cybernetics, Rockville, MD) on a custom-built computer (Digital Storm, Fremont, CA). Specifically, the images were deconvolved using 10 iterations with settings of medium noise, and a blind point spread function. Following deconvolution, images were visually inspected to ensure that striping, ringing, or discontinuity artifacts were not introduced. Further, each image was visually examined to ensure that the borders of puncta were sharpened by the deconvolution and that the membrane staining remained continuous. Following, images were imported into Imaris 7.4.1 (Bitplane AG, South Windsor, CT) for quantification. Though deconvolution improved the visual quality of the images, it was dispensable for accurate quantification of Brp-Short or Dα7 puncta.

The 'Spots' function was used to quantify Brp-Short or Dα7 puncta in Imaris. A region of interest was drawn around the glomerulus being quantified. For individually labeled glomeruli, the standard Imaris box was sufficient to mark the region of interest. Where multiple glomeruli were labeled and a single glomerulus was quantified, a freehand region of interest was drawn in each section using the neuropil staining as a guideline and a 3D region extrapolated by Imaris. The spot size (puncta diameter) was set at 0.6 μm for Brp-Short and 0.8 μm for Dα7, as determined by direct measurement of images and previous studies (*Kremer et al., 2010*). Background subtraction was selected. Following automatic detection of puncta, the threshold was manually adjusted (usually within 5% among different samples) with visual inspection to ensure that all puncta were identified, minimal background staining or noise was recognized, and that individual puncta were not double-counted. In all, the resultant count was recorded as the number of puncta. The accuracy was assessed by comparing puncta detection figures to glomeruli that were counted by hand following 3D projection. In all cases, there was less than 5% difference, supporting the accuracy of the semi-automatic quantification method. In male *Or67d-GAL4* samples that did not express the synaptic label (normally have ~1500 puncta), but were processed identically using primary and secondary antibodies, the number of puncta recognized by this method was <30, suggesting that the 'false positive' rate was low. To ensure that user-introduced noise in threshold adjustment did not introduce significant bias, each image was quantified a second time, at a later time, without access to the prior quantification. If the difference exceeded 5%, the sample was discarded. Further, if the averages of an entire genotype were significantly different from each other (as determined by student's *t* test), the entire cohort was discarded and the experiment repeated.

For the nearest neighbor distance (NND) analysis, the 'DistMin' was calculated in Imaris using the 'Spots to Spots Closest Distance XTension' plug-in for Matlab. Cumulative frequency histograms were compiled in Prism 6.04 (GraphPad Software, San Diego, CA). As each puncta is ~0.6 μm in diameter, the minimum NND should be 0.6. Indeed, few measurements were seen below 0.6. Thus, 'clustering' was defined as puncta with an NND of less than 25% of the diameter. The '% Cluster' was calculated by expressing the number of puncta with an NND value ≤0.75 divided by the total number of puncta.

To obtain neurite volume based on mCD8-GFP or mtdT labeling, the 'Surfaces' function was used. Regions of interest were drawn as above. Background subtraction was set to 3 μm and the smoothing to 0.2 μm; these settings optimally reflected actual neurite labeling. Automatic detection with a 10e5 threshold was then used to remove background, detecting only the glomerulus in question. For smaller objects (MARCM clones), a 10e1 threshold was used and only the specific objects that directly corresponded to mCD8-GFP labeling were selected and counted. Volume was then calculated by Imaris. Density was then expressed as the number of Brp-Short puncta divided by the neurite volume. For images involving a neurite co-stain, the Brp-Short-mStraw channel was masked in Imaris so that only the puncta within the neurite were identified and counted (not background puncta resulting from secondary antibodies, or image noise). Images were processed using Photoshop CS4 (Adobe, San Jose, CA) and levels or contrast adjusted identically for all cases of comparison. For cases of immunofluorescence comparison, brains were stained in parallel, imaged on the same slide in the same session, and processed identically. Statistical and graphical analyses were completed in Excel (Microsoft, Redmond, WA), Prism 6.04 (GraphPad Software, San Diego, CA), and Matlab (Mathworks, Natick, MA).

## Electron microscopy

Adult brains were dissected in 0.1 M cacodylate buffer and fixed at 4°C for 3 hr in 2.5% paraformaldehyde, 5% glutaraldehyde (vol/vol), and 0.06% picric acid (vol/vol) in 0.1 M cacodylate buffer. They were then washed three times for 20 min each in 0.1 M cacodylate buffer on ice and post-fixed with 2% osmium tetroxide for 1 hr at 4°C. Dehydration, infiltration with resin, and staining with uranyl acetate were all

done according to standard protocols. Antennal lobe regions were identified in thick section following 1% toluidine blue in 1% sodium borate staining and ultrathin sections taken from those regions. Sections were imaged on a JEM-1400 (JEOL, Tokyo, Japan) transmission electron microscope at X3,000 to X20,000 magnification. For immuno-EM, adult brains were dissected in 0.1 M sodium phosphate buffer and fixed in modified Zamboni's fixative (4% paraformaldehyde, 1.6% glutaraldehyde, 0.2% saturated picric acid in 0.1 M sodium phosphate buffer at pH 7.4) as described (*Kolodziejczyk et al., 2008*). Brains were then washed in PBS, processed similarly as above, and sectioned. Sections were then blocked in PBST containing 0.05% Tween-20, 0.5% (wt/vol) ovalbumin, and 0.5% (wt/vol) BSA. Sections were incubated overnight at 4°C in rabbit anti-GFP at 1:300 (MBL International, Woburn, MA). Following, grids were washed in PBST and incubated in 5 nm gold-conjugated goat anti-rabbit secondary antibody (Ted Pella, Redding, CA) at 1:50 for 1 hr at RT. Grids were then washed in PBST, incubated in 8% glutaraldehyde for 15 min, washed again in water, and contrast stained with 2% uranyl acetate followed by lead citrate. Ultrastructural analysis was done as previously described (*Watts et al., 2004*; *Mosca et al., 2012*) and T-bar defects quantified using consecutive sections.

## Statistical analyses

For pairwise comparisons, student's *t* test was used to assess statistical significance. For multiple comparisons, a one-way ANOVA was done and corrected for multiple comparisons with posthoc Tukey's multiple comparisons tests done between all genotypes. Analysis was done in Prism 6.0.4 (Graphpad Software, San Diego, CA).

For synapse contribution by individual ORNs, GraphPad Prism was used to fit individual Gaussian relationships to each separate peak for the ORN MARCM data (*Figure 2C–D*, colored dashed lines). The Curve Fitting Tool in Matlab (Mathworks, Natick, MA) was used to determine the best-fit curves that corresponded to the aggregate data set (*Figure 2C–D*, solid black line). In each case, multiple strong candidate relationships that recognized the first four peaks were identified using $r^2$ and root mean square error (RMSE) values. To determine which candidate best fit the data, they were then compared using two *posthoc* tests: corrected Akaike's Information Criteria (AICc) and the *F*-test. AICc is advantageous as it penalizes distributions with more parameters, avoiding potential overfitting, and is useful with binned data. The *F*-test was used to test specific null hypotheses between the models.

To examine the distribution of the individual peaks (i.e., the 'quantal' nature), the standard deviation in the distances between the peaks (S) was calculated. To determine the probability of such an S value occurring, the aggregate DL4 and DM6 data were individually fit to a Poisson distribution or a Gaussian distribution using Matlab. Matlab was then used to generate 50,000 simulations selecting four peaks that fit those distributions. Also, a uniform probability distribution was used to select four peaks between 0 and the upper bound of the DL4 and DM6 data sets, respectively. In all, 300,000 simulations were conducted, and the number of events resulting in an S value less than that observed used to determine a probability of obtaining a more ordered set of four peaks than the observed data.

## Genotypes

In some cases, *UAS-mCD8-GFP* or *UAS-3xHA-mtdT* was included in the genotype (but not shown in the image) to ensure an equivalent number of constructs between control and mutant or rescue conditions.

*Figure 1*: (B–D) *w; UAS-Brp-Short-mStraw/UAS-mCD8-GFP; Or67d-GAL4 [Nr-1]/+; +*. (H–I) DL4 and DM6 = *w; AM29-GAL4/UAS-Brp-Short-mStraw, UAS-mCD8-GFP; +; +*. DA1 = *w; UAS-Brp-Short-mStraw/UAS-mCD8-GFP; Or67d-GAL4 [Nr-1]/+; +*. VA1d = *w, Or88a-GAL4/+ or Y; UAS-mCD8-GFP, UAS-Brp-Short-mStraw/+; +; +*. VA1lm = *w; Or47b-GAL4/UAS-Brp-Short-mStraw, UAS-mCD8-GFP; +; +*.

*Figure 1—figure supplement 1*: (A) *w; UAS-Brp-Short-mStraw/+; Or67d-GAL4 [Nr-1]/UAS-DSyd1-EGFP; +*. (B) *w, UAS-Dcr2; UAS-mCD8-GFP/UAS-Brp-Short-mStraw; Or67d-GAL4 [Nr-1]/+; +*. (C) *w; UAS-mCD8-GFP, UAS-Brp-Short-mStraw/UAS-Brp-IR-v104422; Or67d-GAL4 [Nr-1]/+; +*. (D) *w, pebbled-GAL4/+; +; UAS-Brp-Short-GFP/+; +*.

*Figure 1—figure supplement 2*: (A) *w, Or88a-GAL4/+ or Y; UAS-Brp-Short-mStraw/+; +; +*. (B) *w; Or47b-GAL4/UAS-Brp-Short-mStraw; +; +*. (C and D) *w; AM29-GAL4/UAS-Brp-Short-mStraw; +; +*.

*Figure 2*: (A–F) *hsFLP[122], UAS-mCD8-GFP; AM29-GAL4/UAS-Brp-Short-mStraw; FRT 2A/FRT2A, tubP-GAL80; +*.

*Figure 2—figure supplement 1*: (A–L) *hsFLP[122], UAS-mCD8-GFP; AM29-GAL4/UAS-Brp-Short-mStraw; FRT 2A/FRT2A, tubP-GAL80; +.*

*Figure 3*: (A) *w; UAS-Brp-Short-mStraw/+; Or67d-GAL4 [Nr-1]/+; +.* (B) *w; Mz19-GAL4/UAS-Brp-Short-mStraw; +; +.* (C) *w; UAS-Brp-Short-mStraw/+; NP3056-GAL4/+; +.* (D, G) *w; UAS-Brp-Short-mStraw/UAS-mCD8-GFP; Or67d-GAL4 [Nr-1]/+; +.* (E, H) *w; Mz19-GAL4, UAS-mCD8-GFP/UAS-Brp-Short-mStraw; +; +.* (F, I) *w; UAS-Brp-Short-mStraw/UAS-mCD8-GFP; NP3056-GAL4/+; +.*

*Figure 3—figure supplement 1*: (A) *w; UAS-Brp-Short-mStraw, UAS-mCD8-GFP/+; Or67d-GAL4 [Nr-1]/+; +.* (B) *w; Mz19-GAL4/UAS-Brp-Short-mStraw, UAS-mCD8-GFP; +; +.* (C) *w; UAS-Brp-Short-mStraw, UAS-mCD8-GFP/+; NP3056-GAL4/+; +.*

*Figure 4*: (A) *w; Or47b-GAL4/UAS-Brp-Short-mStraw; UAS-mCD8-GFP/+; +.* (B) *w, Df (X) ten-a; Or47b-GAL4/UAS-Brp-Short-mStraw; +; +.* (C) *w, Df (X) ten-a; Or47b-GAL4/UAS-Brp-Short-mStraw; UAS-Ten-a/+; +.* (D–F) *Control* as in (A). *ten-a* as in (B). *ten-a ORN RNAi = w, UAS-Dcr2; Or47b-GAL4/UAS-Brp-Short-mStraw; UAS-ten-a-IR-v32482/+; +. ten-a +* Ten-a ORN as in C. *ten-m ORN RNAi = w, UAS-Dcr2; Or47b-GAL4/UAS-Brp-Short-mStraw; UAS-ten-m-RNAi-V51173/+; +. ten-a +* Ten-a PN *= w, Df (X) ten-a; Or47b-GAL4/UAS-Brp-Short-mStraw; GH146-QF, QUAS-mCD8-GFP/QUAS-Ten-a; +.*

*Figure 4—figure supplement 1*: (A) *w; Or47b-GAL4/UAS-Brp-Short-mStraw; UAS-mCD8-GFP/+; +.* (B) *w, UAS-Dcr2; Or47b-GAL4/UAS-Brp-Short-mStraw; UAS-ten-m-RNAi-V51173/+; +.* (C) *w, UAS-Dcr2; Or47b-GAL4/UAS-Brp-Short-mStraw; UAS-ten-a-IR-v32482/+; +.* (D) *w, Df (X) ten-a; Or47b-GAL4/UAS-Brp-Short-mStraw; GH146-QF, QUAS-mCD8-GFP/QUAS-Ten-a; +.*

*Figure 4—figure supplement 2*: (A–C) *w; UAS-Brp-Short-mStraw, UAS-mCD8-GFP/AM29-GAL4; +; +.* (D–F) *w, Df (X) ten-a; UAS-Brp-Short-mStraw, UAS-mCD8-GFP/AM29-GAL4; +; +.*

*Figure 5*: Control = *Canton S. ten-a =* Df (X) ten-a; +; +; +.

*Figure 6*: (A) *Canton S.* (B) *Df (X) ten-a; +; +; +.* (C) *w; Or47b-GAL4, UAS-Brp-Short-mStraw/UAS-mCD8-GFP; UAS-mCD8-GFP/+; +.* (D) *w; Or47b-GAL4, UAS-Brp-Short-mStraw/UAS-α-spectrin-dsRNA; UAS-α-spectrin-dsRNA/+; +.* (E) *w; Or47b-GAL4, UAS-Brp-Short-mStraw/UAS-β-spectrin-dsRNA; UAS-β-spectrin-dsRNA/+; +.* (F) Control as in (A) and *ten-a* as in (B). (G) Control as in (C), α-spec ORN RNAi as in (D), β-spec ORN RNAi as in (E), *ten-a = w, Df (X) ten-a; Or47b-GAL4/UAS-Brp-Short-mStraw; +; +. ten-a +* α-spec RNAi *= w, Df (X) ten-a; Or47b-GAL4, UAS-Brp-Short-mStraw/UAS-α-spectrin-dsRNA; UAS-α-spectrin-dsRNA/+; +.*

*Figure 6—figure supplement 1*: (A) *Canton S.* (B) *Df (X) ten-a; +; +; +.* (C) *w; UAS-Brp-Short-mStraw, UAS-mCD8-GFP/+; Or67d-GAL4 [Nr-1], UAS-mCD8-GFP/+; +.* (D) *w; UAS-Brp-Short-mStraw/UAS-α-spectrin-dsRNA; Or67d-GAL4 [Nr-1]/UAS-α-spectrin-dsRNA; +.* (E) *w; UAS-Brp-Short-mStraw/UAS-β-spectrin-dsRNA; Or67d-GAL4 [Nr-1]/UAS-β-spectrin-dsRNA; +.* (F) Genotypes as in (C), (D), and (E).

*Figure 7*: (B, E–F) *w; Mz19-GAL4, UAS-3xHA-mtdT/+; UAS-Dα7-EGFP/+; +.* (C) *w; Mz19-GAL4/+; UAS-Dα7-GFP/+; +.* (D) *w, Or67d-QF; Mz19-GAL4/+; UAS-Dα7-GFP/QUAS-Brp-Short-mStraw; +.*

*Figure 8*: (A) *w, UAS-Dcr2; Mz19-GAL4, UAS-3xHA-mtdT/+; UAS-Dα7-GFP/+; +.* (B) *w, Df (X) ten-a; Mz19-GAL4, UAS-3xHA-mtdT/+; UAS-Dα7-GFP/+; +.* (C) *w, UAS-Dcr2; Mz19-GAL4, UAS-3xHA-mtdT/+, UAS-Dα7-GFP/UAS-ten-m-IR; +.* (D–F) Control as in (A). *ten-a* as in (B). *ten-m* PN RNAi as in C. *ten-a −/− + ten-m* PN RNAi *= w, Df (X) ten-a; Mz19-GAL4, UAS-3xHA-mtdT/UAS-Dcr2; UAS-Dα7-GFP/UAS-ten-m-IR; +.*

*Figure 9*: (B) *w, Or67d-QF; +; QUAS-Brp-Short-mStraw/+; +.* (C) *w, Or67d-QF; Mz19-GAL4/+; QUAS-Brp-Short-mStraw/UAS-ten-m-RNAi-V51173; +.* (D) Control (UAS) = *w; UAS-Brp-Short-mStraw/+; Or67d-GAL4 [Nr-1]/+; +.* Other genotypes as in B–C.

## Acknowledgements

We would like to thank D Luginbuhl and J Perrino for expert technical assistance; B de Bivort for expert statistical guidance; S Sigrist, G Davis, R Dubreuil, and V Favaloro for the kind gift of reagents; members of the Luo Lab and S Zosimus for discussions, along with K Shen, K Beier, D Berns, X Gao, C Guenthner, X Wang, X Wang, A Ward, B Weissbourd, and B Wu for critical comments on the manuscript. This work was supported by grants from the National Institutes of Health (NIH; R01-005982 to LL, K99-DC013059 to TM, 5T32-HD007249 to TM) and the Howard Hughes Medical Institute. LL is an investigator of the HHMI.

# Additional information

## Competing interests

LL: Reviewing editor, *eLife*. The other authors declare that no competing interests exist.

## Funding

| Funder | Grant reference number | Author |
| --- | --- | --- |
| National Institutes of Health | R01-005982 | Liqun Luo |
| Howard Hughes Medical Institute | | Liqun Luo |
| National Institutes of Health | K99-DC013059 | Timothy J Mosca |
| National Institutes of Health | 5T32-HD007249 | Timothy J Mosca |

The funders had no role in study design, data collection and interpretation, or the decision to submit the work for publication.

## Author contributions

TJM, Conception and design, Acquisition of data, Analysis and interpretation of data, Drafting or revising the article, Contributed unpublished essential data or reagents; LL, Conception and design, Drafting or revising the article

## Author ORCIDs

Timothy J Mosca, http://orcid.org/0000-0003-3485-0719

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
