## [Decision Letter]

Thank you for sending your work entitled “Teneurins Regulate the Organization of Synapses in the *Drosophila* Olfactory System” for consideration at *eLife.* Your article has been favorably evaluated by a Senior editor and 3 reviewers, of whom Mani Ramaswami, is a member of our Board of Reviewing Editors.

The Reviewing editor and the other reviewers discussed their comments before we reached this decision, and the Reviewing editor has assembled the following comments to help you prepare a revised submission.

The paper represents a valuable step towards developing optical microscopic methods to quantify synapses and synaptic protein clusters in subsets of the neurons the antennal lobe. It also makes some new contributions to understanding principles of OSN connectivity to the antennal lobe, and to understanding the role of teneurins in this class of central synapse. However, there are several details to be added and some new analyses are necessary before the paper can be accepted.

The major issues that should be addressed are:

1) Provide a detailed and rigorous comparison between UAS-Brp-short and endogenous Brp localization and numbers, based on preparations co-stained for both proteins. E.g., in DA1: in OSN, PN and LN terminals.

2) Provide a rigorous statistical treatment of the “quantal analysis” of OSN-presynatic puncta number, and a mathematically based estimate for “quantal size” in each case.

3) Provide extensive details on the methodology in order to allow the reviewers to assess these methods better and, more importantly, allow them to be reproduced. Also, please place the original stacks used here on a server from where they can be freely downloaded. Where specific settings are used for Autoquant or Imaris, please mention them clearly.

The relevant methods include:

a) Parameters used for image acquisition and each stage of processing.

b) Ensuring the absence of image processing artifacts.

c) For colocalization analyses using relatively objective criteria.

---

## [Author Response]

Below, we first summarize three new experiments we have added to this manuscript independent of reviewers’ critiques, which have strengthened our conclusions regarding the role of the Teneurins in central synapse organization.

A) Transsynaptic Teneurin interaction in regulating synapse number: by making a new transgene, *QUAS-Brp-Short-mStraw*, and combining two binary expression systems (GAL4 and QF), we were able to assess the number of olfactory receptor neuron (ORN) active zones while selectively removing *ten-m* from projection neurons (PNs). Our model predicts that perturbing postsynaptic *ten-m* should reduce the number of presynaptic active zones. We found that this was indeed the case. This offers strong evidence for a transsynaptic Teneurin interaction in the CNS for regulating synapse organization, which has not been previously shown. This data is detailed in Figure 9 with reagent validation in Figure 7.

B) Absence of a role for presynaptic *ten-m* in regulating synapse number: RNAi experiments revealed that knocking down *ten-m* (unlike *ten-a*) in presynaptic ORNs resulted in no synaptic phenotype. This is in contrast to previous findings from the NMJ (63) that showed a minor presynaptic role for *ten-m*. This data is now included in Figure 4.

C) Absence of a role for postsynaptic *ten-a* in regulating synapse number: postsynaptic rescue experiments using two binary expression systems (GAL4 and QF) allowed us to label ORN synapses while restoring Ten-a expression to postsynaptic projection neurons in a *ten-a* mutant background. This did not ameliorate the synaptic phenotype caused by the loss of *ten-a*, cementing a presynaptic role for Ten-a in synapse organization. This data is also now included in Figure 4.

Below we offer point-by-point responses.

*1) Provide a detailed and rigorous comparison between UAS-Brp-short and endogenous Brp localization and numbers, based on preparations co-stained for both proteins. E.g., in DA1: in OSN, PN and LN terminals*.

We have performed additional experiments and have expanded upon the quantifications of Brp-Short puncta in ORNs, PNs, and LNs within the DA1 glomerulus. We show that Brp-Short and endogenous Brp labeling colocalize in all three types of neurons. Moreover, we have added quantifications of Brp-Short puncta in all three classes of neurons, as well as of endogenous Brp puncta in DA1. The data strengthen our conclusions and are detailed in Figure 3—figure supplement 1.

*2) Provide a rigorous statistical treatment of the “quantal analysis” of OSN-presynatic puncta number, and a mathematically based estimate for “quantal size” in each case*.

We have conducted an extensive statistical analysis of the ORN MARCM experiments detailed in Figure 2 and in Figure 2–figure supplements 2 and 3. The text now reflects our analysis, including fitting of sum of Gaussian relationships to the data, prediction of single cell contribution, assessment of significance using multiple tests and a resampling analysis to demonstrate the “quantal” nature of the observed data.

*3) Provide extensive details on the methodology in order to allow the reviewers to assess these methods better and, more importantly, allow them to be reproduced. Also, please place the original stacks used here on a server from where they can be freely downloaded. Where specific settings are used for Autoquant or Imaris, please mention them clearly*.

The relevant methods include:

*a) Parameters used for image acquisition and each stage of processing*.

*b) Ensuring the absence of image processing artifacts*.

*c) For colocalization analyses using relatively objective criteria*.

We have expanded the Methods section to address reviewer requests for extensive details regarding the quantification methodology. We are also working with the Stanford Data Repository to make representative raw data and confocal image stacks available on the Luo Lab website, accessible to any readers, and housed by the Stanford Libraries to ensure a high level of reliability in data access (http://web.stanford.edu/group/luolab/ALsynapse.shtml).